# MobileRL: Online Agentic Reinforcement Learning for Mobile GUI Agents

**Yifan Xu**[1*†], **Xiao Liu**[1,2*], **Xinghan Liu**[1†], **Jiaqi Fu**[1†], **Hanchen Zhang**[1†], **Bohao Jing**[2],
**Shudan Zhang**[1†], **Yuting Wang**[2], **Wenyi Zhao**[2], **Yuxiao Dong**[1]

[1] Tsinghua University    [2] Z.AI

## ABSTRACT

Building general-purpose graphical user interface (GUI) agents has become increasingly promising with the progress in vision language models. However, developing effective mobile GUI agents with reinforcement learning (RL) remains challenging due to the heavy-tailed distribution of task difficulty and the inefficiency of large-scale environment sampling. We present an online agentic reinforcement learning framework MobileRL to enhance GUI agents in mobile environments. Its core component is the Difficulty-**ADA**ptive GRPO (AdaGRPO) algorithm. In AdaGRPO, we design difficulty-adaptive positive replay and failure curriculum filtering to adapt the model to different task difficulties. We introduce the shortest-path reward adjustment strategy to reshape rewards concerning the task length in multi-turn agentic tasks. Those strategies jointly stabilize RL training, improve sample efficiency, and generate strong performance across diverse mobile apps and tasks. We apply MobileRL to two open models (Qwen2.5-VL-7B-Instruct and GLM-4.1V-9B-Base). The resultant MobileRL-9B model achieves state-of-the-art results in terms of success rates on both AndroidWorld (80.2%) and AndroidLab (53.6%). The MobileRL is open-sourced at `https://github.com/THUDM/MobileRL`.

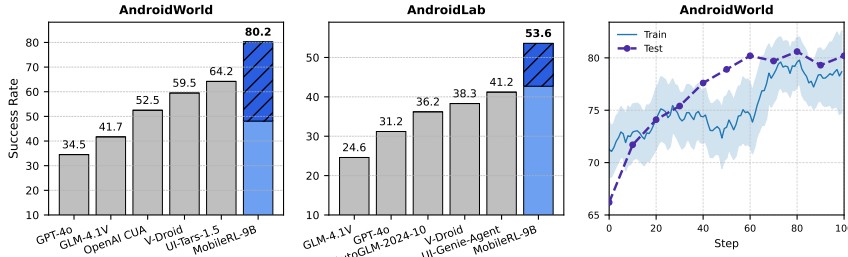

Figure 1: *Left and Center*: Task success rates on AndroidWorld (Rawles et al., 2024) and AndroidLab (Xu et al., 2024); hatched areas indicate gains from MobileRL on top of the SFT model. *Right*: Trajectory-level success rate curves on AndroidWorld train and test sets during RL training.

## 1 INTRODUCTION

GUI agents—powered by vision language models—have enabled zero-shot interaction with web pages and mobile interfaces (Hong et al., 2023; OpenAI; Bai et al., 2025; Liu et al., 2024). To improve them, significant efforts have focused on supervised fine-tuning or offline imitation learning over static expert demonstrations (Rawles et al., 2023; Xu et al., 2024; Bai et al., 2024; Lu et al., 2025). However, these methods suffer from limited behavior coverage and poor error recovery (Chang et al., 2022). Reinforcement learning (RL) with verifiable rewards presents a promising alternative (DeepSeek-AI et al., 2025; Hou et al., 2025). Yet, existing datasets with single-step expert labels (Qin et al., 2025; Luo et al., 2025) are insufficient for training or evaluating policies on agentic tasks (i.e., planning and reasoning over multi-step action sequences). Although early progress has been made on online

---

[*] YX and XL contributed equally.
[†] Work partially done while these authors interned at Z.AI. Corresponding author: YD

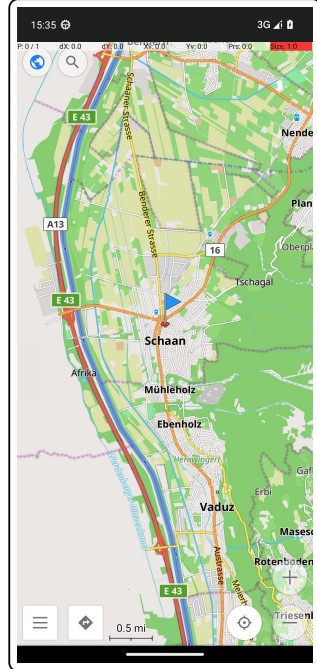 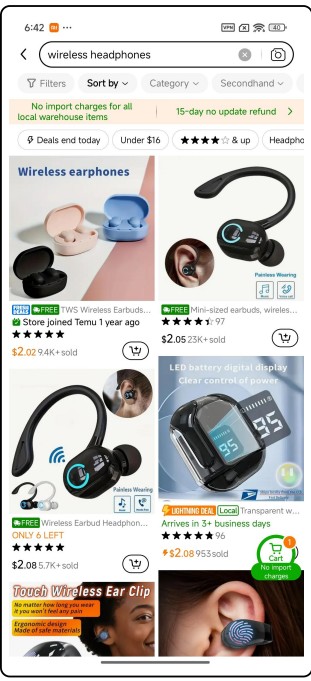 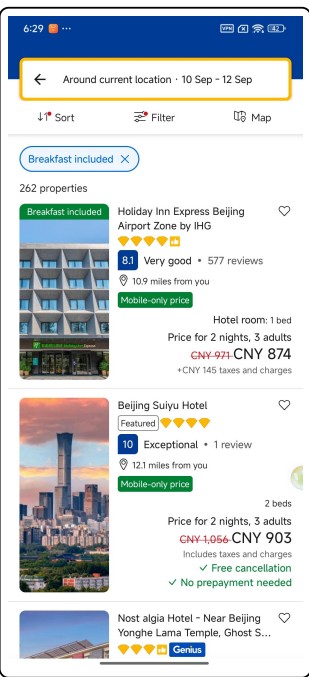

(a) Add a location marker for 47.16, 9.51 in the OsmAnd maps app.

(b) Search for wireless headphones in temu, and sort by price low to high.

(c) Search for hotels on Booking, check-in date is 09-10, check-out date is 09-12, sort by prices.

Figure 2: Example mobile tasks finished by our agent. Our agent can automatically perform tasks according to human instructions in academic benchmarks and real-world applications.

learning for GUI agents (Bai et al., 2024; Dong et al., 2025; Dai et al., 2025), efficiently *scaling agentic RL in interactive mobile simulators* remains largely unexplored.

Specifically, it faces the following technical challenges: (i) *Complex instruction following under sparse positive signals*: base models usually struggle to reliably produce correct action commands for complex, GUI-specific instructions. Due to the heavy cost and latency of mobile emulation, correctly-executed rollouts are rare, resulting in data-inefficient early exploration. (ii) *Large and unstable task difficulty spectrum*: some tasks can succeed with multiple rollouts, while others are persistently unsolvable for the model. Naive sampling wastes computational budget and under-utilizes scarce but informative trajectories (Xu et al., 2024). (iii) *Sampling bottlenecks in large-scale mobile environments*: deploying and managing hundreds of concurrent mobile instances is resource-intensive and hard to reproduce across setups. Low sampling throughput further limits both the scale and efficiency of online agentic RL.

To address these challenges, we present an adaptive online agentic RL framework MOBILERL for advancing mobile GUI agents. MOBILERL consists of three components: reasoning-free supervised fine-tuning (SFT), reasoning SFT, and agentic RL. The two SFT stages provide a warm-up for RL. Specifically, reasoning SFT enhances the handling of long and compositional instructions, reduces costly on-policy trials in mobile simulators, and enables the broad use of open or human-labeled datasets without relying on proprietary models.

To enable effective online agentic RL, we introduce Difficulty–**ADA**ptive Group Relative Policy Optimization (ADAGRPO). Built upon group relative policy optimization (GRPO) (Shao et al., 2024), its core idea is to adapt optimization to instance difficulty and explicitly reward solution efficiency. ADAGRPO designs three key strategies: (i) *Difficulty-Adaptive Positive Replay (AdaPR)* maintains a curated buffer of challenging, high-quality trajectories and balances them with on-policy samples (Mnih et al., 2015; Zha et al., 2019). In sparse-reward mobile environments, difficult successes are rare yet highly informative, emphasizing rare successful rollouts that would otherwise be under-utilized.

(ii) *Failure Curriculum Filtering (FCF)*, as a simplified version of curriculum learning (Matiisen et al., 2019; Narvekar et al., 2020), down-weights persistently unsolvable tasks using online difficulty statistics, reallocating computational budget toward feasible instances. Given the heavy-tailed difficulty distribution observed in mobile agent benchmarks (Xu et al., 2024; Rawles et al., 2024), pruning hard dead-ends improves sample efficiency while retaining signal from recoverable failures. Different from "easy-to-hard" curricula that assume cheap exploration, FCF is resource-aware and aims to avoid repeatedly sampling tasks that remain unsolved under expensive mobile interaction.

(iii) *Shortest-Path Reward Adjustment (SPA)* reshapes the reward function based on completion length, assigning higher returns to shorter solutions. Length-sensitive rewards counteract bias toward verbose and better align with user preferences in mobile interaction contexts. Unlike token-level length penalties used in text-only RLVR, SPA operates on multi-step GUI action trajectories and shapes only successful rollouts, avoiding incentives for premature termination (see Appendix F).Unlike token-level length penalties used in text-only RLVR, SPA operates on multi-step GUI action trajectories and shapes only successful rollouts, avoiding incentives for premature termination.

We implement MOBILERL in a Verl-based framework (Sheng et al., 2024), which supports multi-task, multi-turn agentic RL training. Unlike previous Android simulator implementations (Toyama et al., 2021; Rawles et al., 2024)—which generally do not support true concurrent execution, our framework sustains high throughput that orchestrates *hundreds of Dockerized Android virtual devices (AVDs)* across multiple machines. This setup enables concurrent interaction with over *1,000 environments* while preserving reproducibility. Since most open-source benchmarks and simulators are built upon the Android operating system (Toyama et al., 2021; Rawles et al., 2024), this design ensures seamless compatibility and faithful reproduction of environment behaviors.

We train MOBILERL on Qwen2.5-VL-7B-Instruct (Bai et al., 2025) and GLM-4.1V-9B-Base (Team et al., 2025), producing MobileRL-7B and MobileRL-9B, respectively. MobileRL-9B lifts the success rates to 80.2% on ANDROIDWORLD and 53.6% on ANDROIDLAB, significantly outperforming previous state-of-the-art results (64.2% and 41.2%, respectively). MobileRL-7B outperforms the much larger 72B-parameter models UI-TARS-1.5 (Qin et al., 2025) and UI-GENIE-AGENT (Xiao et al., 2025) (e.g., +16% on ANDROIDWORLD), despite being substantially smaller in scale. Also, extensive ablation studies demonstrate the effectiveness in the design of ADAGRPO.

In summary, our contributions are as follows:

- **MOBILERL Framework & scalable sampling**: We develop MOBILERL with a two-stage warm-up followed by online agentic RL for mobile GUI agents. We further establish a distributed sampling implementation that coordinates *hundreds of Dockerized Android virtual devices (AVDs)*, enabling reproducible large-scale training on Android benchmarks.
- **ADAGRPO Algorithm**: We introduce Difficulty–Adaptive Group Relative Policy Optimization (ADAGRPO), which extends GRPO with (i) *AdaPR* for replaying challenging successful trajectories, (ii) *FCF* for down-weighting persistently unsolved tasks, and (iii) *SPA* for length-sensitive reward shaping, thereby accounting for instance difficulty and solution efficiency.
- **Empirical Results**: Training on Qwen2.5-VL-7B-Instruct and GLM-4.1V-9B-Base yields MOBILERL-7B and MOBILERL-9B. MOBILERL-9B reaches 80.2% on ANDROIDWORLD and 53.6% on ANDROIDLAB, surpassing previous bests (64.2% / 41.2%).

## 2 MOBILERL

We study mobile GUI agents and introduce the MOBILERL framework, as shown in Figure 3, which aims to address three key challenges in interactive mobile environments: (i) following complex instructions under sparse and delayed rewards; (ii) handling a heavy–tailed and unstable task difficulty distribution; and (iii) overcoming large–scale sampling bottlenecks in mobile emulators.

Given a natural–language instruction (e.g., "open the calendar and add an event for tomorrow at 3 pm"), the agent autonomously performs closed–loop interactions with the mobile device. First, it perceives the current screen, grounds UI elements, and executes a sequence of actions without human intervention. The feedback is sparse and it can be observed only upon successful task completion, at which point the interaction terminates or a predefined horizon is reached. The goal is to learn a policy

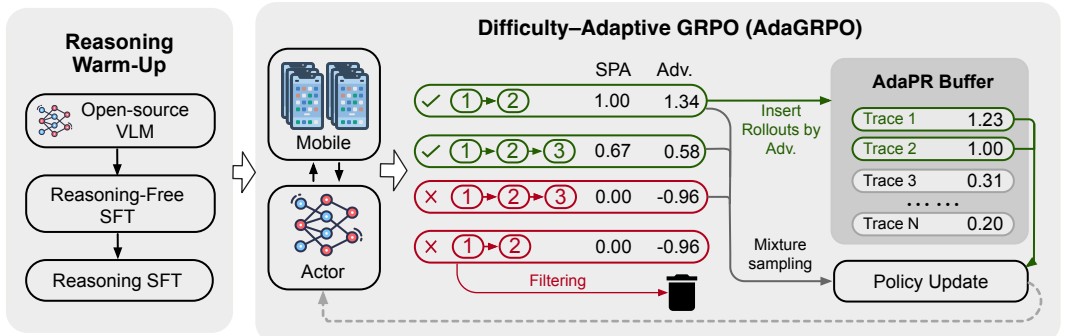

Figure 3: Overview of MOBILERL. It consists of 1) reasoning warm-up with both reasoning-free SFT and reasoning SFT and 2) online agentic RL with ADAGRPO. In ADAGRPO, the warmed-up policy interacts with mobile environments to generate rollouts, which are scored by shortest-path reward adjustment (SPA). High-quality positive trajectories are stored in the AdaPR buffer, while low-performing rollouts are pruned via failure curriculum filtering.

that generates strong performance across applications and tasks, minimizes unnecessary interactions, and maximizes task success.

**Problem Formulation.** We model the mobile GUI agent as a finite-horizon Markov Decision Process (MDP) (Littman, 2009) $\mathcal{M} = (\mathcal{S}, \mathcal{A}, P, r, H, \mu_0)$. The state space $\mathcal{S}$ contains all possible GUI states; a concrete state at time $t$ is $s_t \in \mathcal{S}$, which comprises a screenshot of the device screen together with the structured UI hierarchy parsed from XML metadata (Cf. Appendix C). We also include the model-generated interaction history (i.e., previous `think`/`action` messages) in the state representation, while we do not include the full history of past screenshots/XML due to context-length constraints. The action space $\mathcal{A}$ is a finite set of atomic GUI operations; an action at time $t$ is $a_t \in \mathcal{A}$, including primitives such as `Tap`, `Swipe`, `Type`, `Long Press`, `Launch`, `Home`, `Back`, `Wait`, and a terminating action `Finish`. $P(s_{t+1} \mid s_t, a_t)$ represents the stochastic transition mechanism of the Android OS and installed applications, capturing the uncertainty over possible next states. We consider a finite horizon $H \in \mathbb{N}$. The initial condition is drawn from a joint distribution $\mu_0$ over instruction–state pairs, i.e., $(s_0, c) \sim \mu_0$. Here, $c$ is the natural-language task instruction provided *once* at the beginning of the episode ($t=0$). At each timestep $t = 0, 1, \ldots$, the agent observes only the current state $s_t$ and samples an action according to a policy $\pi_\theta$, i.e., $a_t \sim \pi_\theta(\cdot \mid s_t, c)$. The environment then transitions to $s_{t+1} \sim P(\cdot \mid s_t, a_t)$. An episode yields a trajectory $\tau = \big((s_0, a_0), (s_1, a_1), \ldots, (s_T, a_T)\big)$ and terminates either when the agent intentionally selects `Finish` in a success state or when the horizon is reached, with $T \leq H$. The reward is assigned only after task completion, such that $R(\tau) = r(s_T, a_T)$ with $r(s_T, a_T) \in \{0, 1\}$ indicating success (1) or failure (0). Consequently, learning maximizes the success probability: $\theta^\star = \arg\max_\theta \mathbb{E}_{(s_0, c) \sim \mu_0, \ \tau \sim \pi_\theta}\big[R(\tau)\big]$.

## 2.1 THE MOBILERL FRAMEWORK

To build a powerful mobile use agent, we present the MOBILERL framework. It comprises three components: reasoning-free supervised fine-tuning (SFT) on expert demonstration data, an iterative warm-up stage by reasoning SFT, and agentic RL with a difficulty–adaptive policy optimization strategy we developed in this work.

**Reasoning-Free SFT.** In agentic RL training, sampling in virtual-device environments is usually inefficient; thus, starting online RL directly from a base model was found to be excessively time-consuming. Therefore, we perform SFT with expert demonstration data obtained by following the data collection protocol of (Xu et al., 2024), supplemented with the training split of the publicly available AndroidControl dataset (Li et al., 2024). Note that this data is reasoning-free.

**Reasoning SFT.** To further construct a stronger reasoning policy initializer, we perform reasoning SFT via an iterative reasoning refinement strategy over the expert dataset. Manually collected

expert demonstrations dataset for mobile use often contains only the final action sequence, omitting intermediate reasoning. Training solely on such "black-box" trajectories yields opaque policies. We leverage an off-the-shelf instruction model to bootstrap a reasoning-augmented training set from raw demonstrations, yielding a structured and transparent policy initialization. Concretely, we iteratively build the reasoning instruction–tuning pairs in three stages:

- Bootstrap sampling: For each task $x$ with expert answer $a^*$, the Instruct model $M$ generates diverse candidate reasoning–action pairs $(c_k, a_k)$. Whenever $a_k = a^*$, we retain $(x, c_k, a^*)$ in $\mathcal{D}_R$.
- Supervised fine-tuning: Train an initial reasoning policy $\pi_0^R$ on $\mathcal{D}_R$.
- Iterative refinement: At iteration $t$, $\pi_t^R$ proposes candidates; those matching $a^*$ are scored by correctness. The best explanation $c^*$ is added to $\mathcal{D}_{\text{new}}$, and $\pi_{t+1}^R$ is obtained by fine-tuning.

The resulting reasoning-oriented fine-tuning corpus is trained for two epochs to produce the reasoning warm-up model used for agentic RL training.

**Agentic RL.** During agentic RL (multi-turn) training, we face the challenges of immediate reward assignment and sampling efficiency, which are discussed and addressed by building upon the group relative policy optimization (GRPO) (Shao et al., 2024).

Briefly, GRPO advances proximal policy optimization (Schulman et al., 2017) by replacing the learned value baseline with an on-the-fly, group-relative baseline computed from a set of trajectories for the same task. Given a shared initial condition $(s_0, c) \sim \mu_0$, we sample a group of $G$ trajectories $\mathcal{G} = \{\tau_1, \dots, \tau_G\}$ by rolling out $\pi_{\theta_{\text{sample}}}$. Let $T_i$ denote the number of steps for trajectory $\tau_i$, $s_{i,t}, a_{i,t}$ be the state and action at step $t$, and the trajectory-level reward for $\tau_i$ be $R(\tau_i) \in \{0, 1\}$. We define the group-relative trajectory-level advantage for any step $(s_{i,t}, a_{i,t})$ on trajectory $\tau_i$ as $\hat{A}_{i,t} \equiv \frac{R(\tau_i) - \mu_R}{\sigma_R}$, where $\mu_R$ and $\sigma_R$ are the mean and standard deviation of $\{R(\tau_j)\}_{j=1}^G$. The GRPO loss can be written in empirical form as

$$\mathcal{L}^{\text{GRPO}}(\theta) = -\frac{1}{G} \sum_{i=1}^{G} \frac{1}{T_i} \sum_{t=1}^{T_i} \min\Big( \rho_{i,t}(\theta)\, \hat{A}_{i,t},\; \text{clip}\big(\rho_{i,t}(\theta), 1 - \epsilon, 1 + \epsilon\big)\, \hat{A}_{i,t} \Big) \qquad (1)$$

where $\rho_{i,t}(\theta) = \frac{\pi_\theta(a_{i,t} \mid s_{i,t})}{\pi_{\theta_{\text{sample}}}(a_{i,t} \mid s_{i,t})}$ is the token-wise importance sampling (IS) ratio. We add the KL loss $\beta\, D_{\text{KL}}\big(\pi_\theta \,\|\, \pi_{\theta_{\text{ref}}}\big)$ to prevent the model from deviating too much from the prior distribution.

## 2.2 DIFFICULTY–ADAPTIVE GRPO

We develop Difficulty–Adaptive GRPO (ADAGRPO) with three strategies—shortest-path reward adjustment (SPA), difficulty–adaptive positive replay (AdaPR), and failure curriculum filtering (FCF)—to address the challenges faced in mobile agentic RL.

First, in multi-turn mobile agentic tasks—where immediate rewards are absent within a single round, unlike single-turn settings (Lu et al., 2025; Team et al., 2025)—the reward allocation strategy must be redesigned. Beyond assigning a uniform terminal reward, we introduce *Shortest-Path Reward Adjustment (SPA)*, which reshapes rewards with respect to task length. The goal of the adjustment is to provide more informative learning signals, guiding the model toward accurate and efficient completion paths and facilitating the computation of trajectory-level advantages.

Second, a further challenge arises from the uniform sampling strategy employed in standard GRPO. In mobile use scenarios—where each sample carries a significant computational cost—this approach results in poor sample efficiency, particularly due to the repeated inclusion of inherently unsolvable tasks. To mitigate this, we adapt data collection and training based on instance difficulty through two mechanisms: Difficulty–Adaptive Positive Replay (AdaPR) and Failure Curriculum Filtering (FCF). At the same time, we restrict redundant successful trajectories to avoid unnecessary updates and promote training efficiency.

### 2.2.1 SHORTEST-PATH REWARD ADJUSTMENT (SPA)

In mobile tasks, the environment returns a binary terminal reward $r \in \{0, 1\}$ (Xu et al., 2024; Rawles et al., 2024) indicating task success. Previous approaches typically broadcast this reward to every

timestep, i.e., $R(s_t, a_t) = r,\ t = 0, \ldots, T$, so that the per-step signal remains aligned with the sparse objective. However, assigning identical rewards to all successful rollouts biases training toward *longer* trajectories, since they contribute more gradient terms. To counteract this, we introduce SPA, which re-scales the reward for each trajectory $\tau_i$ as

$$R^{\text{SPA}}(s_t, a_t) = r(\tau_i) \left( 1 - \alpha \frac{T_i - T_{\min}}{T_i} \right), \qquad T_{\min} = \min_{\tau_j \in \mathcal{T}_{\text{succ}}} |\tau_j|,\ \alpha \in (0, 1]. \qquad (2)$$

where $T_i = |\tau_i|$ is the length of trajectory $\tau_i$, and $\mathcal{T}_{\text{succ}} = \{\tau_j \mid r(\tau_j) = 1\}$ denotes the set of *successful* trajectories for the current problem instance. Here $T_{\min}$ is the length of the shortest successful trajectory in $\mathcal{T}_{\text{succ}}$, and $\alpha \in (0, 1]$ controls the penalty strength. In this formulation, shorter sequences are not automatically considered better; unsuccessful early terminations still receive a reward of $0$. This adjustment encourages the policy to prefer shorter, successful paths without sacrificing the success rate. As in prior work, we broadcast $R^{\text{SPA}}(s_t, a_t)$ to every timestep of the trajectory and use it to compute group-relative advantages.

### 2.2.2 DIFFICULTY–ADAPTIVE POSITIVE REPLAY (ADAPR)

In sparse-reward mobile environments, successful yet challenging rollouts are rare but highly informative; effectively leveraging them enhances the learning signal and stabilizes policy updates. Therefore, inspired by the paradigm of experience replay (Mnih et al., 2015; Zha et al., 2019), we introduce difficulty-adaptive positive replay (AdaPR) strategically to retain and reuse challenging, high-value trajectories while blending them with fresh on-policy samples. We introduce key components of AdaPR: buffer construction for high-quality trajectory selection and mixture sampling to balance replay and exploration.

**Buffer Construction.** At iteration $t$, the rollout set is $\mathcal{T}_t = \{\tau^{(1)}, \ldots, \tau^{(N)}\}$, collected under the current policy $\pi_{\theta_t}$. We compute the trajectory-level advantage and insert the top $\kappa$ trajectories into the replay buffer $\mathcal{B}$.

**Mixture Sampling.** Each policy update is performed on a mini-batch of $M$ trajectories obtained from the mixture distribution $q(\tau) = \gamma\, p_{\mathcal{B}}(\tau) + (1 - \gamma)\, p_{\text{on}}(\tau)$ where $p_{\text{on}}$ is the on-policy distribution $\pi_{\theta_t}$ and $p_{\mathcal{B}}$ is the empirical distribution over $\mathcal{B}$. To keep the replay contribution under control, at most $\gamma M$ trajectories with the highest current advantage are drawn from $\mathcal{B}$, preserving on-policy diversity. We provide a quantitative analysis of the replay buffer effectiveness and the token-level PPO-style clip ratio in Appendix B.

### 2.2.3 FAILURE CURRICULUM FILTERING (FCF)

To avoid repeatedly sampling tasks that yield zero reward—which wastes computation and hinders the collection of positive advantage data, we propose failure curriculum filtering. In FCF, any task producing all-zero rewards for two consecutive epochs enters a three-epoch cooldown, during which its sampling probability is reduced according to $w_{\text{task}} = \exp(-f)$, where $f$ is the number of consecutive failure epochs; after this cooldown period, the task is permanently removed. This method can be regarded as a variant of curriculum sampling (Matiisen et al., 2019; Narvekar et al., 2020). To avoid excessive hyperparameter tuning, we simplify it to target only the exclusion of the most difficult subset of data. Tasks with consistently high failure counts are permanently removed from the sampling pool. For stability, failure histories from previous training are retained. Notably, FCF is applied only to the training sampling distribution; all evaluations are conducted on the full test set without removing any tasks. Notably, FCF is applied only to the training sampling distribution; all evaluations are conducted on the full test set without removing any tasks.

**Summary.** In summary, MOBILERL consists of reasoning-free SFT, reasoning SFT, and difficulty-adaptive RL for training mobile GUI agents. Reasoning-free SFT helps build a strong action foundation from expert demonstrations, while reasoning SFT adds intermediate reasoning to improve instruction following and policy transparency. On top of this initialization, agentic RL with ADA-GRPO addresses the challenges of sparse terminal rewards, heavy-tailed task difficulty, and expensive sampling. Specifically, SPA reshapes terminal rewards for denser feedback, AdaPR strategically reuses challenging successful trajectories, and FCF filters out persistently-unsolvable tasks.

Table 1: Success rates (%) of proprietary and open-source models on AndroidWorld and AndroidLab for mobile GUI interaction tasks.

| Models | #Params | AndroidWorld | AndroidLab |
|---|---|---|---|
| *Proprietary Models* | | | |
| GPT-4o-2024-11-20 (OpenAI, 2023) | - | 34.5 | 31.2 |
| Claude-Sonnet-4-20250514-thinking (Anthropic, 2023) | - | 41.0 | 40.6 |
| UI-Tars-1.5 (Qin et al., 2025) | - | 64.2 | 38.3 |
| AUTOGLM-2024-10 (Liu et al., 2024) | - | – | 36.2 |
| *Open Models* | | | |
| Qwen2.5-VL-7B-Instruct (Bai et al., 2025) | 7B | 27.6 | 10.1 |
| GLM-4.1V-9B-Thinking (Team et al., 2025) | 9B | 41.7 | 24.6 |
| UI-Tars-7B (Qin et al., 2025) | 7B | 33.0 | 32.6 |
| V-Droid (Dai et al., 2025) | 8B | 59.5 | 38.3 |
| UI-Genie-Agent (Xiao et al., 2025) | 72B | - | 41.2 |
| MOBILERL *(Ours)* | | | |
| w/ Qwen2.5-VL-7B | 7B | 72.0 | 42.5 |
| w/ GLM-4.1V-9B-Base | 9B | **80.2** | **53.6** |

## 3 EXPERIMENTS

### 3.1 EXPERIMENTS SETTINGS

**Datasets and Benchmarks.** In the two-stage SFT process, we utilize data from human annotations and AndroidControl (Li et al., 2024), constructing 97.9k and 23.6k training steps, respectively. During the RL stage, for the final MobileRL-9B setting, we employ an interactive training set from AndroidWorld (Rawles et al., 2024), consisting of 2,000 tasks with verifiable rewards. We evaluate on two interactive Mobile benchmarks: AndroidWorld, which includes 116 tasks across 20 apps, and AndroidLab, which comprises 138 tasks across 9 apps. Both benchmarks provide interactive environments. We further assess performance on the static dataset AndroidControl (Li et al., 2024), which contains 8,444 test samples. Due to space limitations, additional details on the training and test datasets are provided in Appendix D.

**Baselines.** Our baselines encompass both closed- and open-source agents and models. The closed-source LLMs include GPT-4o-2024-11-20 (OpenAI, 2023) and Claude-Sonnet-4-20250514-thinking (Anthropic, 2023), and closed-source agents UI-Tars-1.5 (Qin et al., 2025) and Auto-GLM (Liu et al., 2024). The open-source VLMs, including Qwen2.5-VL-7B-Instruct (Bai et al., 2025), GLM-4.1V-9B-Thinking (Team et al., 2025), UI-Tars-7B (Qin et al., 2025), V-Droid (Dai et al., 2025) and UI-Genie-Agent (Xiao et al., 2025), are used (see Appendix E.3 for input modality and evaluation protocol details).

### 3.2 MAIN RESULTS

We evaluate MOBILERL by using Qwen2.5-VL-7B and GLM-4.1V-9B-Base as backbones on online interactive benchmarks, AndroidWorld and AndroidLab. As shown in Table 1, MOBILERL significantly outperforms previous results. With Qwen2.5-VL-7B as the backbone, MOBILERL achieves 72.0% on AndroidWorld and 42.5% on AndroidLab, outperforming previous state-of-the-art methods. By using GLM-4.1V-9B as the backbone, the performance of MOBILERL further improves to 80.2% on AndroidWorld and 53.6% on AndroidLab, achieving the best results among the compared baselines under our evaluation protocol.

### 3.3 ABLATION STUDY

To evaluate the contributions of the MOBILERL framework and the components of the ADAGRPO algorithm, we start with two base models and progressively apply Reasoning-Free SFT, Reasoning

| Models | AndroidWorld | AndroidLab |
|---|---|---|
| Qwen2.5-VL-7B-Instruct (Bai et al., 2025) | 27.6 | 10.1 |
|   + Reasoning-Free SFT | $50.2_{+22.6}$ | $36.9_{+26.8}$ |
|     + Reasoning SFT | $56.8_{+6.6}$ | $38.7_{+1.8}$ |
|       + ADAGRPO (MOBILERL-7B) | $72.0_{+15.2}$ | $42.5_{+3.8}$ |
| GLM-4.1V-9B-Base (Team et al., 2025) | 7.7 | 10.1 |
|   + Reasoning-Free SFT | $48.1_{+40.4}$ | $42.7_{+32.6}$ |
|     + Reasoning SFT | $66.2_{+18.1}$ | $45.0_{+2.3}$ |
|       + ADAGRPO (MOBILERL-9B) | $80.2_{+14.0}$ | $53.6_{+8.6}$ |

(a) Improvements of task success rates by incrementally applying Reasoning-Free SFT, Reasoning SFT, and ADAGRPO to the base models, respectively.

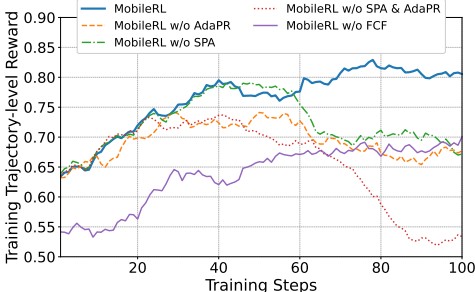

| Models | AndroidWorld |
|---|---|
| MOBILERL | **71.1** |
|   w/o AdaPR | 63.6 |
|   w/o SPA | 69.1 |
|   w/o AdaPR&SPA | 58.5 |
|   w/o FCF | 64.8 |
|   w/o ADAGRPO | 56.8 |

(b) Training trajectory-level rewards from the Android-World environment with respect to training steps.

(c) Test performance on the Android-World test set under different variants.

Figure 4: Ablation studies of the MOBILERL framework and its ADAGRPO algorithm. We use the Reasoning SFT model with Qwen2.5-VL-7B-Instruct backbone for the ablation of the ADAGRPO algorithm. All test set results are averaged over three runs to mitigate the impact of randomness.

SFT, and ADAGRPO. Then, using the Qwen2.5-VL-7B-Instruct model trained with Reasoning SFT as the initialization point and on the AndroidWorld training set, we conduct an analysis of the impact of each component of ADAGRPO—AdaPR, SPA, and FCF.

**MOBILERL Ablation.** We summarize stage-wise gains in success rate (SR) in Table 4a. For Qwen2.5-VL-7B, the combined improvements of MOBILERL are +44.4% on AndroidWorld and +32.4% on AndroidLab; For GLM-4.1V-9B, the overall gains are +72.5% and +43.5%. Overall, Reasoning-Free SFT delivers a strong initial lift, and Reasoning SFT offers additional improvements. Building upon the strong foundation established by the preceding stages, the ADAGRPO stage further augments the final performance, achieving an improvement exceeding 10% on the AndroidWorld dataset and over 5% on the AndroidLab test set.

**ADAGRPO Ablation.** The design of ADAGRPO covers SPA, AdaPR, and FCF. The ablations are performed on four settings: (i) MOBILERL w/o AdaPR (no replay), (ii) MOBILERL w/o SPA (no reward shaping), (iii) MOBILERL w/o AdaPR & SPA (neither), and (iv) MOBILERL w/o FCF (uniform sampling; this variant keeps both SPA and AdaPR enabled).

We report on-policy trajectory reward curves during training (excluding replayed trajectories) in Figure 4b and the success rates on the AndroidWorld in Table 4c. The curve reports the binary verifiable success reward (before applying the SPA length penalty), so it reflects whether each on-policy rollout succeeds on the sampled training task. The curve reports the binary verifiable success reward (before applying the SPA length penalty), so it reflects whether each on-policy rollout succeeds on the sampled training task. To avoid bias from the AndroidLab reward model, we use only AndroidWorld in this study. Each component of ADAGRPO contributes to improving the performance of MOBILERL. Specifically, we have the following observations:

- **FCF under constraints.** With a 100-step budget ($> 40$ hours), FCF plays a key role in filtering. Removing it biases early sampling toward overly hard tasks, yielding many negatives and a lower

Table 2: Results on the AndroidControl (Li et al., 2024). Note that data from AndroidControl was not included during the RL stage.

| Model | AndroidControl-Low | | | AndroidControl-High | | |
|---|---|---|---|---|---|---|
| | Type | Grounding | SR | Type | Grounding | SR |
| UI-Tars-7B | 98.0 | 89.3 | 90.8 | 83.7 | 80.5 | 72.5 |
| UI-Tars-72B | 98.1 | 90.8 | 91.3 | 85.2 | 81.5 | 74.7 |
| UI-Genie-7B | 98.1 | 89.3 | 94.3 | 83.5 | 82.9 | 74.5 |
| UI-Genie-72B | 98.3 | 95.4 | 94.8 | 84.9 | **86.3** | 77.0 |
| MOBILERL-*7B* | | | | | | |
| w/o RL | 96.7 | 93.7 | 91.9 | 87.0 | 73.6 | 72.8 |
| w/ RL | 96.9 | 92.4 | 91.4 | 86.3 | 71.5 | 71.3 |
| MOBILERL-*9B* | | | | | | |
| w/o RL | **98.6** | **95.8** | **95.9** | **89.8** | 78.1 | **77.5** |
| w/ RL | 97.9 | 94.6 | 94.3 | 87.0 | 74.5 | 73.9 |

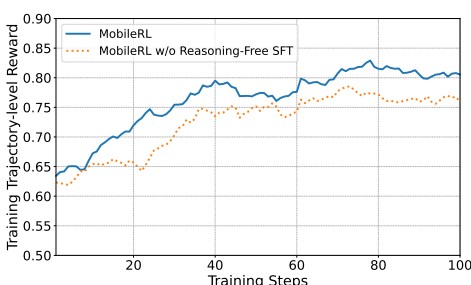

Table 3: Effect of reasoning-free SFT evaluated on AndroidWorld.

reward ceiling. The w/o FCF curve keeps rising, suggesting stronger results with more efficient pipelines or simulators.

- **FCF only (w/o AdaPR & SPA).** Training is initially stable but collapses after about 30 steps, indicating that AdaPR and SPA are necessary for stabilizing the training.
- **Effect of AdaPR.** After about seven steps (once the replay buffer is populated), the gap between w/o AdaPR and the full MOBILERL method grows, showing the benefit of replay.
- **Effect of SPA.** Noticeable gains of SPA appear after roughly 60 steps, likely because the lack of step-wise control leads to overly long trajectories.

**Is Reasoning-Free SFT Still Necessary?** We apply supervised fine-tuning on the expert dataset without reasoning traces, which we term *Reasoning-Free SFT*. This raises a key question: *Is fine-tuning with expert data that lacks reasoning still beneficial?*

We compare MOBILERL (Reasoning-Free SFT + Reasoning SFT + ADAGRPO) with MOBILERL without Reasoning-Free SFT in Figure 3. Interestingly, our experiments show that incorporating Reasoning-Free SFT consistently improves performance on AndroidWorld. It suggests that without explicit reasoning traces, (Reasoning-Free) SFT contributes to enhancing final results for MOBILERL.

**Performance on Offline Dataset.** As shown in Appendix D, Reasoning-Free and Reasoning SFT stages include training data from AndroidControl, which has been converted into MobileRL format. In Table 2, we present the results of MobileRL on the AndroidControl test set. The SR score of MobileRL-9B w/o RL version surpasses all previous models, achieving the state of the art, while the RL version largely maintains the score.

**Success Rates by Task Complexity.** We divide the AndroidWorld test set by rounded-up *Complexity* (Rawles et al., 2024): complexity level 1 ($\leq$10 steps), level 2 (11–20), level 3 (21–30), and level 4+ ($>$30). We run eight test trials at temperature 1.0 and report pass@1/2/4/8 in Figure 5. MOBILERL yields consistent gains at all complexity levels. Notably, post-RL pass@1 exceeds pre-RL pass@8, indicating substantial effectiveness. Consistent with AdaPR's design for heavy-tailed difficulty, larger gains occur on high-complexity tasks. For a more fine-grained, multi-dimensional difficulty analysis beyond step-based complexity, see Appendix E.2.

**Impact of SPA on Step Efficiency.** Although SPA has the smallest impact on overall accuracy in the ablation, its effect on step efficiency is clear. As shown in Figure 6, partitioning tasks by complexity reveals that MOBILERL with SPA consistently completes tasks in fewer steps across all difficulty levels. Moreover, when we compare cases where both models are correct (BC), both are wrong (BW), and those where only one is correct, MOBILERL with SPA more frequently yields shorter trajectories in every group.

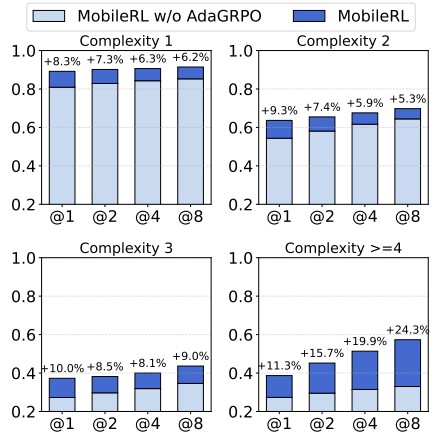

Figure 5: Pass@$k$ on AndroidWorld by task complexity (Rawles et al. (2024)) levels. Pass@$k$ is the fraction of tasks solved within the top-$k$ attempts.

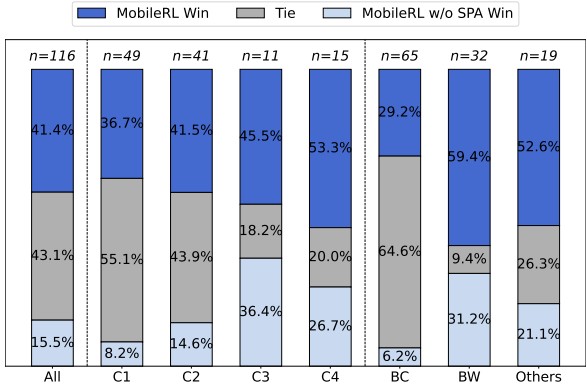

Figure 6: Win rate of MOBILERL vs. MOBILERL w/o SPA, where a *win* means completing a task with fewer steps. $n$ denotes the number of task templates per category. Categories: *All* (all templates); *C1–C4* (complexity levels 1–4); *BC/BW* (both methods correct/wrong); *Others* (exactly one method correct).

## 4 RELATED WORK

**Mobile GUI Agents.** Recent work leverages powerful language models to build agents that operate real PCs and phones (Agashe et al., 2025; Qin et al., 2025; Lai et al., 2025b), including Android agents that perceive GUIs and act via taps, swipes, and text (Toyama et al., 2021; Xu et al., 2024). To improve action prediction and learning, frameworks explore multimodal exploration (Yang et al., 2023), modular reasoning (Lai et al., 2025a), verifier-driven control (Dai et al., 2025), and small-LM code-based execution (Wen et al., 2025). Yet many systems still rely on offline RL or single-turn data: DigiRL uses offline demonstrations (Bai et al., 2024); U1-R1 trains on single-step episodes (Lu et al., 2025); and UI-Tars applies DPO in an offline regime (Qin et al., 2025), leaving online, multi-turn RL for adaptive mobile agents unexplored. Studies emphasize realistic applications: AppAgent (Yang et al., 2023) evaluates closed-source models on real-world apps, while A3 (Chai et al., 2025) offers a realistic app suite with an autonomous evaluation protocol that reduces human effort.

**Benchmarks for Mobile Agents.** Benchmarking generally follows two tracks. Static or replay-based settings—AndroidControl (Li et al., 2024), Android in the Wild (Rawles et al., 2023), MobileAgentBench (Wang et al., 2024a), and Mobile-Bench (Deng et al., 2024) offer plenty of tasks and trajectories. These benchmarks fall short for real-world evaluation, as fixed trajectories and screens limit agents' ability to handle uncertainty and exploration. Interactive emulator environments—AndroidWorld (Rawles et al., 2024), AndroidLab (Xu et al., 2024), and B-MOCA (Lee et al., 2024)—span diverse apps and realistic tasks, yet remain challenging for current agents. Current mobile GUI benchmarks are limited: they lack asynchronous, parallel VM interaction for scalable training. To the best of our knowledge, all public mobile GUI benchmarks to date target the Android operating system.

## 5 CONCLUSION

In this work, we present MOBILERL, an agentic RL framework that advances mobile GUI agents. It achieves this by combining staged initialization with an adaptive reinforcement learning algorithm (ADAGRPO). Training begins with reasoning-free SFT on large-scale expert demonstrations, followed by a reasoning SFT stage that adds intermediate rationales and reduces cold-start exploration costs. Building on this, we introduce Difficulty–**ADA**ptive GRPO (ADAGRPO), which enhances GRPO with shortest-path reward adjustment, adaptive positive replay, and failure curriculum filtering. These three strategies improve sample efficiency and guide policies toward more accurate and efficient task completion. Experiments on AndroidWorld and AndroidLab demonstrate that MOBILERL with open models significantly outperforms both open-source and proprietary baselines.

ACKNOWLEDGMENT

This work was supported by NSFC 62495063 and 62276148, Z251100008425007, BNR2026RC01009.

STATEMENT

**Ethics Statement**   This work involves human annotation for collecting human interaction data. Details of the data collection process and privacy protection measures are discussed in Appendix D.2. We ensure that no other components of our work pose potential risks or harms.

**Reproducibility Statement**   We provide detailed discussions of all training and evaluation datasets in Appendix D, including their sources, preprocessing steps, deduplication methods, and evaluation protocols. Training algorithms, code frameworks, parameter settings, and implementation details are described in Appendix B.

To facilitate reproducibility of the sampling framework and experimental results, we release an anonymous repository at `https://anonymous.4open.science/r/MobileRL-iclr-4513`. The repository includes instructions for configuring the sampling framework, with most complex dependencies (e.g., sampling framework workers and AVDs) packaged into Docker images, ensuring that users can launch rollouts on Linux machines. The reproduction procedures for experimental results are also documented in the repository.

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

## A   CASE STUDY

We present a case study from AndroidWorld evaluating three agents: the Reasoning-Free SFT agent, Reasoning SFT agent, and MOBILERL agent. We choose `SimpleCalendarAddRepeatingEvent`, which requires creating a recurring event titled "Review session for Budget Planning," starting on 2023-10-15 at 14:00, lasting 60 minutes, repeating weekly without end, and including the description: "We will understand business objectives. Remember to confirm attendance."

All agents successfully configured the title and basic settings in the initial steps. Their performance diverged in subsequent steps. The Reasoning-Free SFT agent made timing errors (setting 16:00 instead of 15:00) and executed redundant checks, revealing weak task understanding (Figure 7). The Reasoning SFT agent skipped an adjustment step, yielding an incorrect event duration (Figure 8). By contrast, the full MOBILERL agent completed the task accurately and efficiently, satisfying all requirements without redundant operations (Figure 9).

## B   TRAINING DETAILS

### B.1   HYPERPARAMETERS

In both the Reasoning-Free SFT and Reasoning SFT stages, we conducted fine-tuning for two epochs, employing a cosine-decayed learning rate scheduled from $1 \times 10^{-5}$ to $1 \times 10^{-6}$. The training was implemented using `Llama-Factory`, with packing mode enabled to accelerate training efficiency. All images were used at their full resolution without compression.

In the RL stage, we extend the GRPO framework from `Verl` with our customized functionalities. The main hyperparameters are summarized in Table 4.

### B.2   ACTION SPACE.

We design a series of actions based on AndroidLab (Xu et al., 2024), which supports a concise set of actions for GUI interaction, as described in Table 5.

### B.3   IMPLEMENT DETAILS

#### B.3.1   PRUNING NEGATIVE TRAJECTORIES

To stabilize training, we prune trajectories with the lowest advantages, reducing noisy samples in the replay buffer. High-advantage trajectories, if stored, are expected to be sampled only once on average. Thus, we maintain a maximum positive-to-negative trajectory ratio of 1:2 by randomly discarding the lowest-advantage trajectories.

We further study a pruning strategy that discards overly frequent erroneous trajectories from the reinforcement learning buffer before each update. As depicted in Figure 10, this filtering keeps the

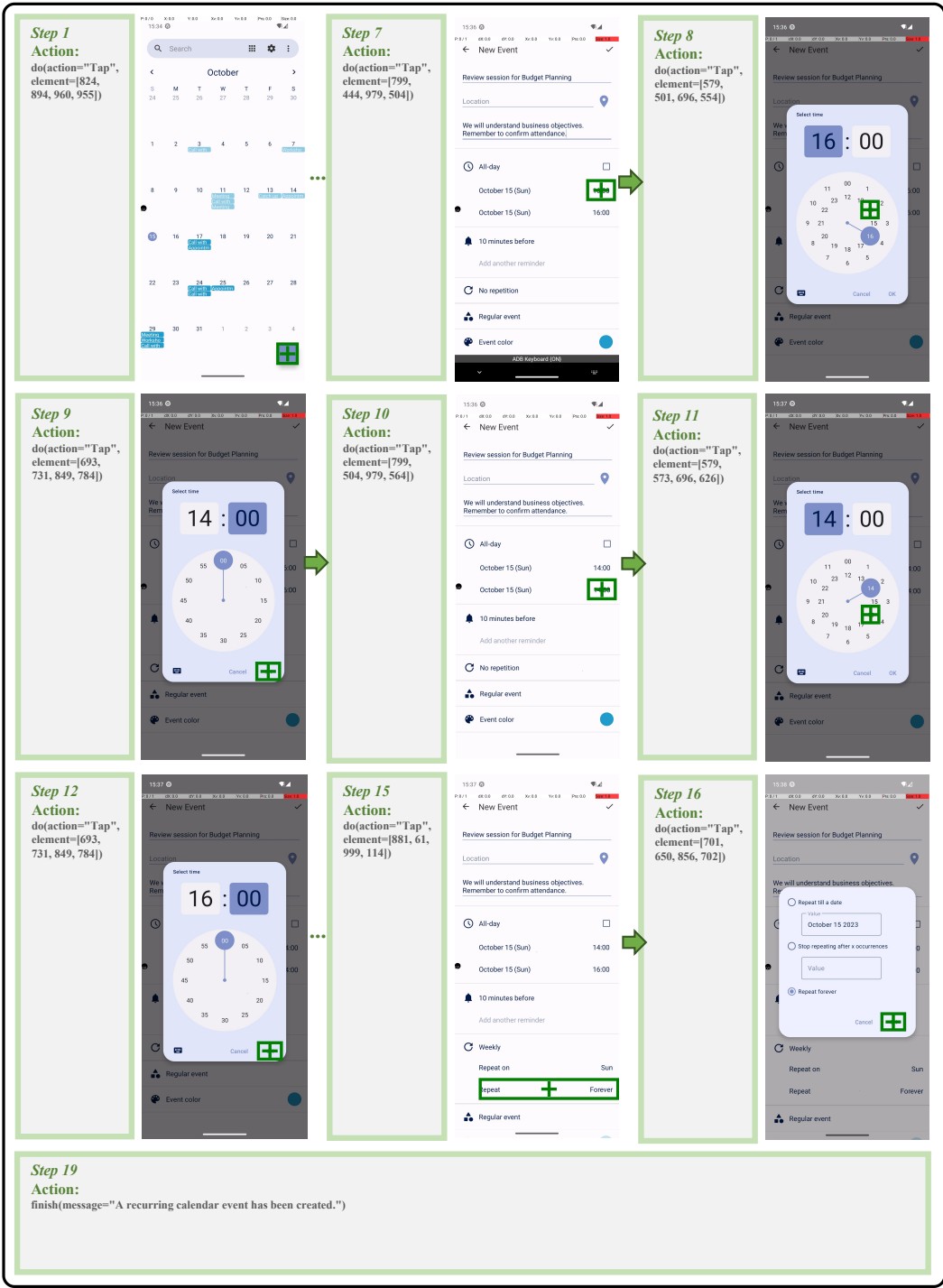

Figure 7: The Reasoning-Free SFT agent, where the expert data does not include the reasoning part, in the `SimpleCalendarAddRepeatingEvent` task in AndroidWorld. Because of space constraints, we only present the most critical and notable steps for comparison.

policy entropy consistently higher during training. By pruning trajectories whose advantages remain persistently negative, the agent avoids being driven by detrimental gradients; probability mass is instead spread over a broader action space, fostering exploration and delaying premature convergence. This pruning strategy ultimately leads to more robust policy learning.

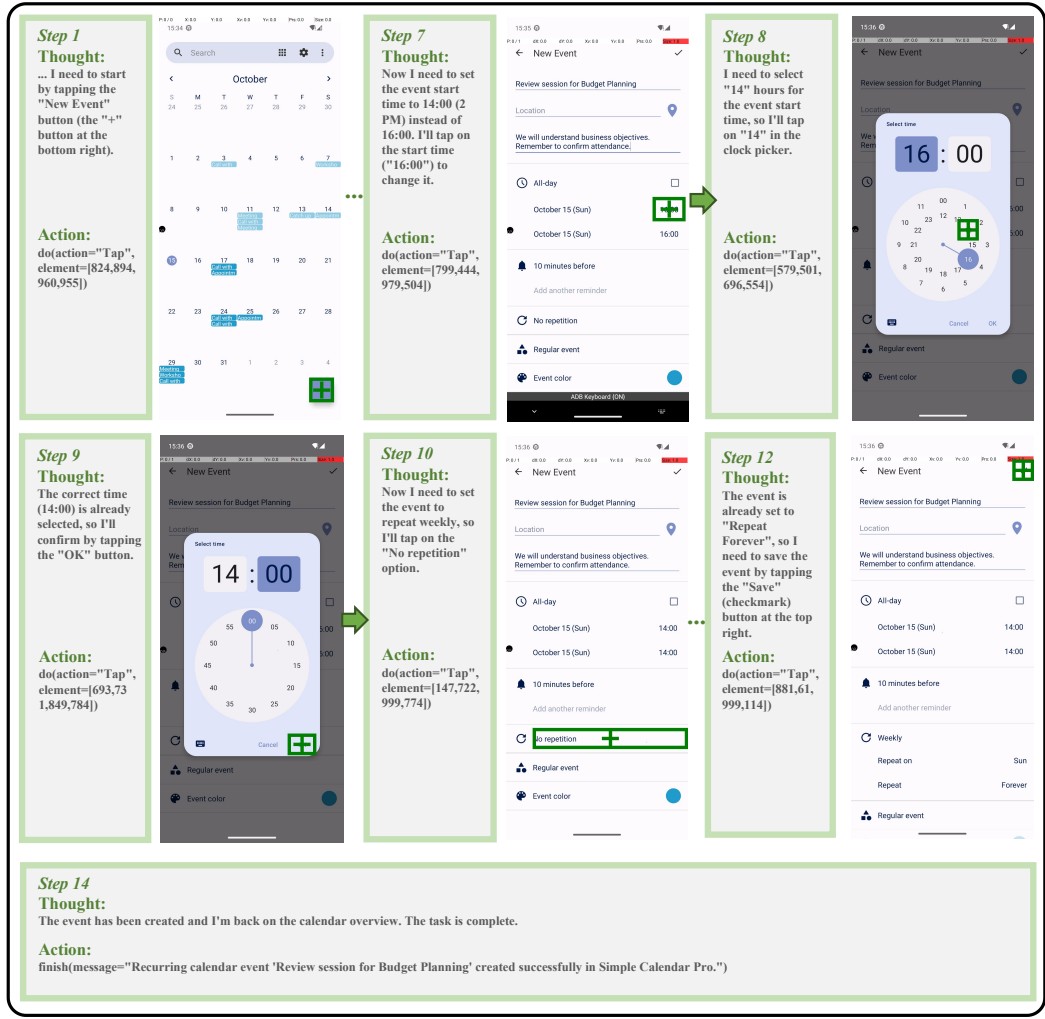

Figure 8: Reasoning SFT agent in the `SimpleCalendarAddRepeatingEvent` task in AndroidWorld. Because of space constraints, we only present the most critical and notable steps for comparison.

### B.3.2 INFLUENCE OF IMAGE RESOLUTION

In our preliminary experiments, we applied a uniform compression to images (maximum pixels per image = 500,000) across all three stages in order to accelerate training, yielding the initial results. The standard practice, however, is to use uncompressed images. For experiments based on Qwen2.5-VL-Instruct, we observed that this difference did not significantly affect performance while greatly accelerating training. In contrast, for experiments based on GLM-4.1V-9B-Base, we found notable accuracy differences between compressed and uncompressed settings, as shown in Table 6. Therefore, we updated the reported results to those obtained with full-resolution images.

This change in accuracy is reasonable in the SFT stage, since the AndroidLab tasks contain more question-answering oriented tasks, where higher resolution images often make it easier to read screen content. In the RL stage, clearer images allow the agent to explore more thoroughly, leading to higher scores.

### B.3.3 EFFECTIVENESS OF ADAPR BUFFER

AdaGRPO employs a mixture sampling strategy that incorporates recently successful trajectories stored in a replay buffer. Since these trajectories originate from earlier policy snapshots, their

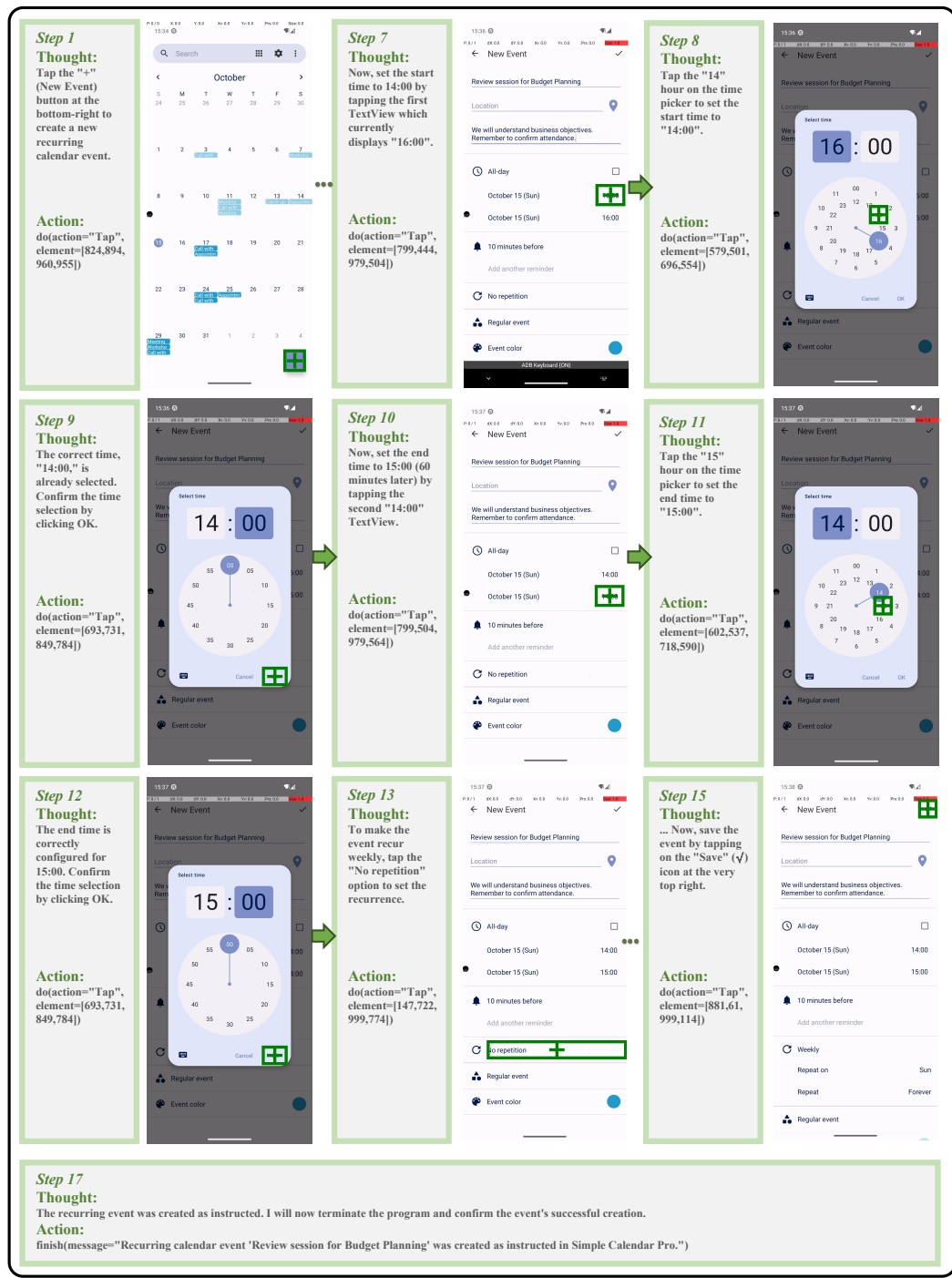

Figure 9: MOBILERL in the `SimpleCalendarAddRepeatingEvent` task in AndroidWorld. Because of space constraints, we only present the most critical and notable steps for comparison.

distribution may deviate from the current policy, raising the possibility that they fall outside the PPO-style clipping range used in the GRPO objective and therefore contribute marginally to the gradient. To ensure that replayed trajectories remain sufficiently close to the on-policy distribution, we enforce strict entry and exit criteria for the AdaPR buffer and conduct targeted empirical analyses. These analyses demonstrate that stored trajectories stay near the current policy, maintain a low clip ratio, and provide an effective gradient signal throughout training.

Table 4: Summary of Main Hyperparameters

| Component | Hyperparameter | Value |
|---|---|---|
| Data | Max Prompt Length | 16384 |
| Data | Max Response Length | 4096 |
| Data | Train Batch Size | 256 |
| Data | Validation Batch Size | 256 |
| Actor / Policy | Strategy (Parallelism) | FSDP |
| Actor / Policy | PPO Micro Batch Size/GPU | 4 |
| Actor / Policy | Learning Rate (LR) | 1e-6 |
| Actor / Policy | Gradient Clipping | 1.0 |
| Actor / Policy | Clip Ratio | 0.2 |
| Actor / Policy | PPO Epochs | 1 |
| Rollout & Sampling | Sampling Temperature | 1.0 |
| Rollout & Sampling | Max New Tokens | 4096 |
| Rollout & Sampling | Number of Samples (n) | 16 |
| Rollout & Sampling | Max Turns | 50 |
| Rollout & Sampling | Max Pixels | 5000000 |
| Rollout & Sampling | Min Pixels | 65536 |
| Algorithm | KL Loss Coefficient/ $\beta$ | 0.001 |
| Algorithm | SPA/ $\alpha$ | 1.0 |
| Algorithm | ADAPR/ Replay Buffer Size | 256 |
| Algorithm | ADAPR/ $\gamma$ | 1.0 |
| Algorithm | ADAPR/ $\kappa$ | 0.25 |

Table 5: Action Space for Mobile GUI Interaction. We utilize a merging action space from Android-Lab (Xu et al., 2024) and AndroidControl Li et al. (2024), which represents screen positions with bounding boxes aligned to XML data.

| Action | Parameters | Description |
|---|---|---|
| Tap | `element=[x1,y1,x2,y2]` | Tap at the rectangle defined by top-left (x1,y1) and bottom-right (x2,y2). |
| Type | `text={string}` | Enter the given string into the focused input field. |
| Swipe | `direction={up/down/left/right}` `dist={short/medium/long}` `element=[x1,y1,x2,y2] (optional)` | Swipe in the given direction over the specified distance. Optionally constrain to the rectangle element. |
| Long Press | `element=[x1,y1,x2,y2]` | Press and hold on the given rectangle area. |
| Launch | `app={AppName}` | Launch the named application. |
| Back | none | Press the system Back button. |
| Home | none | Press the system Home button. |
| Wait | none | Wait for three seconds. |
| Finish | `message={string} (optional)` | End the session with an optional message. |

**Buffer entry and exit conditions**  During training, only the top-$\kappa$ trajectories (sorted by estimated advantage), with $\kappa = 25\%$ in all experiments, are eligible to enter the AdaPR buffer. Since the success rate increases throughout training, this truncation ensures that nearly all stored trajectories correspond to valid task completions (i.e., correct think–act sequences), preventing drift toward outdated or incorrect behavior.

Trajectories leave the buffer following two strictly enforced rules:

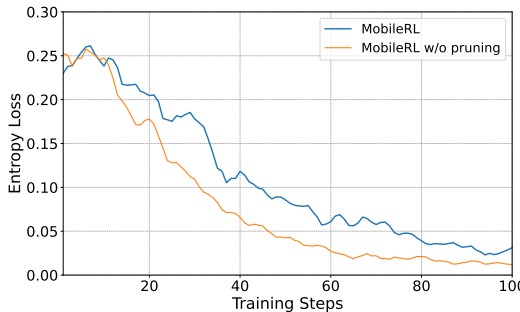

Figure 10: Effect of the pruning-negative strategy evaluated on AndroidWorld.

Table 6: Comparison of success rate between compressed and original images across different training stages and datasets.

| Model | Compressed Images | | Original Images | |
|---|---|---|---|---|
| | AndroidWorld | AndroidLab | AndroidWorld | AndroidLab |
| Reasoning-Free SFT | 48.9 | 39.8 | 48.1 | 42.7 |
| Reasoning SFT | 66.1 | 40.3 | 66.2 | 45.0 |
| ADAGRPO (MobileRL-9B) | 75.8 | 46.8 | 80.2 | 53.6 |

1. **Age-based eviction:** Any trajectory older than $N = 1/\kappa$ rounds (i.e., $N = 4$ with our settings) is immediately removed.

2. **Capacity-based eviction:** If the buffer exceeds the maximum capacity, the least advantageous trajectories are removed first (descending advantage order).

These constraints guarantee that all replayed trajectories originate from *recent* policy snapshots whose divergence from the current policy remains limited.

**Empirical analysis of clipping behavior**    To quantify the degree to which replay-induced off-policy samples fall into the clipping range, we report the fraction of tokens for which the importance ratio exceeds the clipping threshold. Figure 11 plots the token-level clip ratio throughout training. Before enabling AdaPR, the clip ratio fluctuates between 0.5%–1%. After enabling AdaPR, the ratio momentarily rises to approximately 6% early in training as the replay buffer populates, and stabilizes at 3%–4% afterward. Considering that the batch mixture is approximately 1:1 between on-policy and replayed samples, this implies that over 90% of tokens originating from replay remain within the unclipped region and thus contribute fully to the gradient.

**On-policy vs. off-policy task coverage**    In Figure 12, we show the fraction of unique tasks covered by on-policy rollouts and off-policy (replayed) trajectories. After the initial steps, the ratio remains stably around 1:1 throughout training, demonstrating that the buffer effectively preserves task diversity and does not collapse to a narrow subset of tasks.

**Effect on trajectory-level accuracy**    Finally, Figure 13 directly compares the trajectory-level success rate between training with vs. without replay. Across the entire training horizon, including replay, improves trajectory-level correctness by more than 5% absolute, further validating that replayed samples provide meaningful and beneficial gradient signal.

**Summary**    Together, these findings show that replayed successful trajectories are only mildly off-policy due to strict buffer management, remain largely within clipping bounds, and contribute substantially to learning. This demonstrates that AdaGRPO maintains high sample efficiency while effectively exploiting historical successful rollouts.

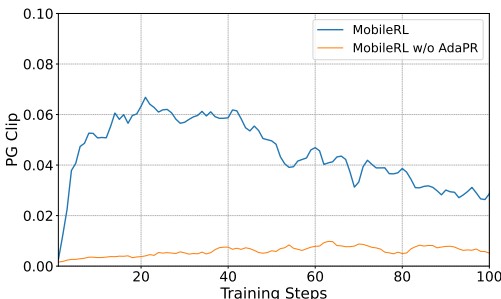

Figure 11: **Token-level clip ratio during training.** Enabling AdaPR temporarily increases the fraction of clipped tokens as the buffer warms up, but the ratio stabilizes at 3%–4%, indicating that over 90% of buffer-originating tokens remain inside the clipping region and contribute fully to the GRPO objective.

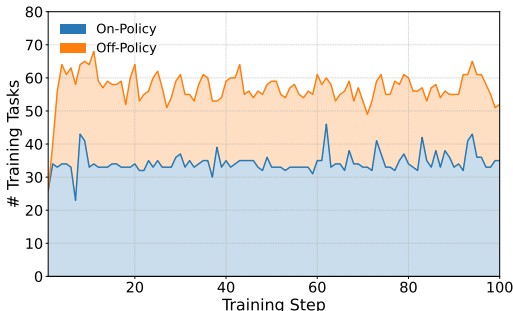

Figure 12: **On-policy vs. off-policy task coverage.** After warm-up, the replay buffer and on-policy rollouts cover a comparable number of unique tasks, preserving task diversity throughout training.

## C   XML Preprocessing for UI Representation

The original XML from the Android accessibility service defines the layout and elements of the user interface, including all components on a page. As a result, it contains many nodes used solely for structural or layout purposes, which do not provide useful semantic information. Moreover, scrollable pages often contain more nodes than are visible on the screen, leading to the inclusion of many off-screen nodes.

### C.1   Removal of Off-Screen Nodes

We first determine whether to retain off-screen nodes via the input parameter `remain_nodes`:

- **`remain_nodes=True`**: Off-screen nodes are preserved, e.g., when summarizing the full page content without requiring scrolling.
- **`remain_nodes=False`**: Off-screen nodes are removed to avoid interference during action simulation (e.g., tapping, scrolling).

In the original XML, a node is considered on-screen if its `bounds` property lies entirely within the screen dimensions $[0, 0]$ to $[Window\_Height, Window\_Width]$ and is contained by its parent node. We check this condition recursively to identify on-screen nodes.

### C.2   Removal of Redundant Nodes

Nodes that do not convey functional or semantic information are removed. A node is considered *functional* if it satisfies at least one of the following:

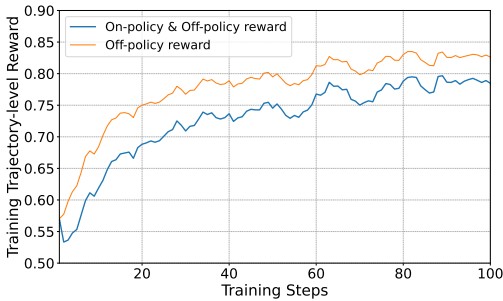

Figure 13: **Trajectory-level success rate with vs. without replay.** Incorporating replay yields more than a 5% absolute improvement in trajectory-level accuracy across training, demonstrating that replayed trajectories provide meaningful and effective gradient signal.

- Any of the boolean attributes is `True`: `checkable`, `checked`, `clickable`, `focusable`, `scrollable`, `long-clickable`, `password`, `selected`.
- The `text` or `content-desc` attribute is non-empty.

All nodes failing these criteria are considered redundant and are deleted.

### C.3 ATTRIBUTE SIMPLIFICATION

Attribute descriptions in the original XML are verbose and token-expensive. We simplify them as follows:

- Keep only `True` values for the boolean functional attributes listed above (omit `False` values).
- Remove `index`, `resource-id`, and `package` (not useful for semantic understanding).
- For `class`, retain only the last component (e.g., `android.widget.FrameLayout` → `FrameLayout`).
- Merge `text` and `content-desc` attributes and display them separately.
- Retain `bounds` in full, as it indicates the node's position on the page.

### C.4 EXAMPLE TRANSFORMATION

Original node:

```
<node index="0" text="Audio Recorder"
      resource-id="com.dimowner.audiorecorder:id/txt_title"
      class="android.widget.TextView"
         package="com.dimowner.audiorecorder"
      content-desc="" checkable="false" checked="false"
         clickable="false"
      enabled="true" focusable="false" focused="false"
         scrollable="false"
      long-clickable="false" password="false" selected="false"
      bounds="[221,1095][858,1222]" />
```

Simplified node:

```
TextView;;Audio Recorder;[221,1095][858,1222]
```

## D DATA COLLECTION

In this section, we present the composition and construction methodology of the training and testing data used at each stage, as well as their respective proportions.

## D.1 Data Distribution

We present the source distribution of the first two stages of SFT data in Table 7. In the RL stage, for the MobileRL-9B setting, we utilized 2,000 tasks from AndroidWorld, and MobileRL-7B setting, we utilized tasks form AndroidWorld and AndroidLab. Details of our data construction, deduplication, and reward acquisition methods are provided in a later section.

Table 7: Data distribution and labels across the two SFT training stages by steps.

| Stage | Android Control (Low) | Android Control (High) | Human Annotation | Total |
|---|---|---|---|---|
| Reasoning-Free SFT | 21.3k | 14.3k | 62.2k | 97.9k |
| Reasoning SFT | 7.2k | 4.1k | 12.2k | 23.6k |

## D.2 Human Annotation

Following the approach in Xu et al. (2024), we collected a portion of human-annotation data and additionally employed model-driven self-exploration on the online interactive environment, performing online rollouts and selecting the correct action sequences for training.

**Privacy Protection** In this work, the data collection process for model training inevitably involves capturing page screenshots and corresponding XML files. To address this, a privacy protection interface that does not retain raw data is applied before storage, ensuring that all sensitive information is filtered out in advance and that only sanitized data is preserved. Building upon this foundation, the overall scope of privacy protection is defined to cover both textual and visual information, with sensitive elements being systematically identified and anonymized. For textual data, the detection process primarily employs regular expression–based rules to recognize personal identifiers such as social media accounts, vehicle registration numbers, shipment tracking codes, order numbers, bank account details, email addresses, passport information, national identification numbers, and phone numbers. Once detected, these elements are replaced with standardized masking tokens that indicate the category of the information without revealing the actual content, thereby preserving semantic integrity while mitigating risks of exposure. For visual data, the privacy protection mechanism extends to the automatic detection of faces, QR codes, and text extracted from images through optical character recognition. Detected sensitive regions within images are then masked or obfuscated, producing new visual outputs in which private content is effectively concealed. Taken together, these measures form a comprehensive privacy-preservation strategy that integrates rule-based textual analysis with multimodal image processing, ensuring robust safeguards against the leakage of personal or identifying information throughout the data collection and annotation pipeline.

## D.3 AndroidLab

### D.3.1 Data Processing

The AndroidLab (Xu et al., 2024) dataset consists of an online evaluation set containing 138 problems, all of which are equipped with verifiable evaluation mechanisms. In our study, we directly adopt the SR (Success Rate) metric provided by AndroidLab as the primary reward indicator. However, since AndroidLab does not include a corresponding verifiable training set, we construct one by means of a model-driven expansion strategy. Specifically, we utilize the applications included in AndroidLab to automatically generate candidate training tasks, which are subsequently subjected to manual verification to ensure both feasibility and non-overlap with the test set. Through this process, we compile a total of 1,103 training problems.

Since ground-truth test methods do not accompany these generated problems, it is not possible to rely on the same direct evaluation mechanism used for the test set. To address this issue, we train a reward model that serves as an auxiliary evaluator of correctness. This reward model is employed in two stages of our training pipeline: (i) during the Reasoning SFT stage, it functions as a filter to ensure the quality and validity of training data; and (ii) during the Reinforcement Learning stage, it provides a learnable reward signal that guides policy optimization.

### D.3.2 Reward Model for AndroidLab

Since the AndroidLab environment does not provide rule-based rewards for training data, we adopt a VLM-based reward model to supply reward signals during reinforcement learning. Specifically, we first execute all training and test tasks multiple times using medium-capability VLMs, including different versions of our models and proprietary VLMs, generating execution traces. Then, strong proprietary VLMs assign binary scores to each trace. For scoring, each model receives the full task screenshot sequence concatenated into a single image, along with step-by-step action descriptions, and is instructed to produce a reasoning process before outputting a score. For training-set tasks (non-rule-based), we use the majority vote among the three scores as the label and retain the reasoning; for test-set tasks, we use the existing rule-based reward as the label and keep only the model scores and reasoning consistent with it. This process yields several thousand labeled samples, which we use to fine-tune GLM-4.1V-9B-Thinking as the base model. After training, we evaluate the reward model on a curated set of 1000 AndroidLab traces with verified labels, selecting the best-performing version (86% accuracy) for online reinforcement learning.

**System Prompt for Trace Evaluation**   The following system prompt guides the VLM in determining whether an agent has successfully completed a task.

```
You are an expert in determining whether a task
has been successfully and completely completed. You will receive:

1. The task description.
2. Step-by-step page states in XML format.
3. The agent's action descriptions.
4. A single image containing screenshots of all steps.
   - Green rectangles and crosses mark tap regions and positions.
   - Green arrows indicate swipe directions.
   - Red text shows typed input.

Action formats and meanings:
- do(action="Tap", element=[x1,y1,x2,y2])
    Tap on the specified screen region; green cross is the tap
        point.
- do(action="Launch", app="xxx")
    Launch the specified app.
- do(action="Type", text="xxx")
    Enter the specified text (shown in red).
- do(action="Swipe", element=[x1,y1,x2,y2], direction="x",
    dist="x")
    Swipe in the indicated direction; green arrow shows the swipe
        path.
- do(action="Long_Pres", element=[x1,y1,x2,y2])
    Long press on the specified region.
- do(action="Back")
    Navigate back to the previous screen.
- finish(message="xxx")
    End the task with the given message.

Scoring rules:
If the task is fully and correctly completed,
output a score of 1; otherwise, 0.

Output format:
<analysis>
Step 1 analysis: <Your analysis>
Step 2 analysis: <Your analysis>
...
Final step analysis: <Your analysis>
</analysis>
```

```
<ans>
[Your score]
</ans>
```

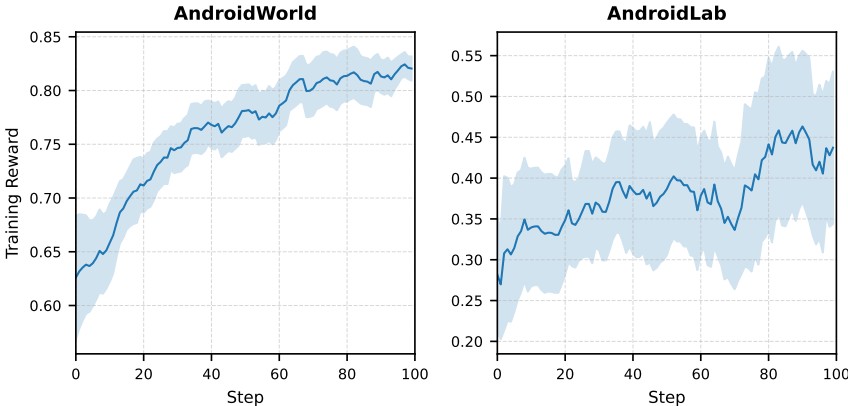

Figure 14: Trajectory-level rewards with 95% CIs on training sets, with the reasoning SFT version of Qwen-2.5-VL-Instruct as backbone model. showing consistent performance growth.

**Analysis**   We compared the training curves of AndroidWorld and AndroidLab (trained simultaneously), as shown in Figure 14. The training curve of AndroidWorld is smoother compared to that of AndroidLab, which we attribute to the use of a rule-based reward in AndroidWorld, whereas AndroidLab relies on a reward model. However, the reward model cannot fully match the accuracy of the rule-based reward. Therefore, we believe that evaluating whether a broader range of tasks has been successfully completed without relying on rule-based rewards remains an important challenge for future work.

## D.4   ANDROIDWORLD

### D.4.1   DATA PROCESSING

The AndroidWorld dataset (Rawles et al., 2024) comprises 116 test problems, each equipped with verifiable reward signals. Since its native action space differs from ours, we adapt the operation code to align with our action space. AndroidWorld further provides randomized initialization parameters, which enable the generation of a large number of training tasks with verifiable rewards by replacing predefined substitutable components in the tasks (e.g., input content) and by varying the initial device states. To prevent overlap with the evaluation set, we remove all tasks whose specific content and initial states coincide with those in the test set. From the remaining pool, we curate 2,000 tasks to serve as the training set.

## D.5   ANDROIDCONTROL

### D.5.1   DATA PROCESSING

Each test case in the AndroidControl (Li et al., 2024) dataset includes `episode_id`, `goal`, `screenshots`, `accessibility_trees`, `screenshot_widths`, `actions`, `screenshot_heights`, and `step_instructions`. The format of input information and output actions differs from that of our action space, and we provide transformation details in Table 8. Certain transformations are required for an evaluation to be performed. The dataset consists of 8,444 test samples, 690 validation samples, and 74,714 training samples.

**Transformation from Accessibility Tree to XML**   We first transform the input information `accessibility_tree` to XML. The XML obtained from mobile pages contains nodes with the following attributes: `index`, `resource-id`, `class`, `package`, `content-desc`, `checkable`, `checked`, `clickable`, `enabled`, `focusable`, `focused`, `scrollable`, `long-clickable`, `password`, `selected`, and `bounds`. We extract and set these corresponding attributes from each node of the accessibility tree. This ensures that the converted XML is

Table 8: The correspondence between the AndroidControl action space and MobileRL action space, as well as the judgment details for each action.

| AC Action | MobileRL Action | Evaluation Details |
|---|---|---|
| click | Tap | Consider correct if the action type matches and the click point falls within the Tap bounding box. |
| long_press | Long Press | Consider correct if the action type matches and the long_press point falls within the Long Press bounding box. |
| scroll | Swipe | Consider correct if the action type matches and the scroll direction corresponds to the Swipe direction. |
| open_app | Launch | Consider correct if the action type matches and the Launch app matches. |
| input_text | Type | Consider correct if the action type matches and the Type text F1 score with the input_text text is higher than 0.5. |
| navigate_home | Home | Consider correct if the action type matches. |
| navigate_back | Back | Consider correct if the action type matches. |
| wait | Wait | Consider correct if the action type matches. |

consistent with the XML obtained directly from mobile pages. We recursively process through steps as follows:

- Recursively process the three levels of `accessibility_tree`, `window`, and `node`.
- Set the corresponding XML attribute for each attribute of the node.
- Compress the converted XML into compressed_XML format.

Here is a transformation example:

Accessibility tree node:

```
nodes {
    bounds_in_screen {
      right: 1080
      bottom: 2400
    }
    class_name: "android.widget.FrameLayout"
    package_name: "com.zoho.meeting"
    text_selection_start: -1
    text_selection_end: -1
    window_id: 1733
    is_enabled: true
    is_visible_to_user: true
    actions {
      id: 4
    }
    actions {
      id: 8
    }
    actions {
      id: 64
    }
    actions {
      id: 16908342
    }
  }
```

XML node:

```
<node index="0" text="" resource-id=""
   class="android.widget.FrameLayout" package="com.zoho.meeting"
   content-desc="" checkable="false" checked="false"
```

| App | Example instruction |
|-----|---------------------|
| Chrome | Open Chrome and switch to incognito browsing mode. |
| Gmail | Compose a new email in Gmail to `alice@example.com` with the subject `Meeting Reminder` and the body `Please do not forget our meeting at 10 AM tomorrow.` |
| GoogleMaps | Search for the nearest hospital in Google Maps. |
| TikTok | Post the latest video from the gallery with the caption `Weekend vibes` in TikTok. |
| temu | Search for a hat in temu and sort the results by price from low to high. |

Table 9: Representative examples from the out-of-domain public-use apps evaluation suite. The complete set of 106 tasks is provided in the online test sheet in the supplementary material.

```
clickable="false" enabled="true" focusable="false"
focused="false" scrollable="false" long-clickable="false"
password="false" selected="false" bounds="[0,0][1080,2400]"/>
```

**Transformation from Point to Bounding Box**   The grounding actions in AndroidControl use points. Since our grounding actions use bounding boxes rather than points, conversion of grounding coordinates is required. We attempt to match all UI boxes on a page. If the point falls within a UI box, the original click coordinates are converted to the coordinates of that UI box.

**Evaluation Details**   Our evaluation metric follows Lian et al. (2025), as it provides open-source complete evaluation code, as shown in Table 8. Notably, since our grounding actions use bounding boxes instead of points, we have modified the judgment logic for positioning operations. A grounding action is considered correct if the point coordinates of the original action fall within the UI box predicted by the model. For the AndroidControl Low evaluation, we only conduct single-turn conversations with the agent. The provided information includes the instruction for the current step, the screenshot, and compressed XML data. For the AndroidControl High evaluation, we maintain a history with the agent and replace incorrect actions. The provided information includes the total task, the screenshot for the current step, and compressed XML data.

# E   ADDITIONAL EXPERIMENT

## E.1   OUT-OF-DOMAIN EVALUATION ON PUBLIC-USE ANDROID APPLICATIONS

To evaluate whether the improvements transfer beyond AndroidWorld/AndroidLab-style environments, we construct an out-of-domain evaluation suite on public-use Android applications. This suite measures the extent to which a trained policy generalizes to previously unseen apps and user goals that are outside the academic benchmarks.

**Applications and task coverage.**   We include **21** widely used apps spanning system utilities, productivity, navigation, e-commerce, and social media: *Settings*; *Booking*, *Expedia*; *Chrome*; *Contacts*, *Gmail*, *Google Drive*, *GoogleDocs*, *GoogleKeep*, *GoogleCalendar*; *GoogleMaps*, *GooglePlayBooks*, *GooglePlayStore*; *Joplin*, *OsmAnd*; *Quora*, *Reddit*; *TikTok*, *Twitter (X)*; *temu*, *Broccoli*. For each app, we manually design 4–7 realistic natural-language tasks, leading to a total of 106 tasks (e.g., 9 for *Chrome*, 6 for *Gmail*, 6 for *TikTok*, and 7 for *temu*).

**Task examples.**   The tasks are formulated as natural-language instructions that reflect realistic user goals, such as connecting to a specified WiFi network, composing and sending emails, performing web searches in incognito mode, or interacting with social media and shopping apps. Table 9 shows representative examples.

**Annotation protocol and scoring.**   For each task, the agent executes the instruction in the corresponding app. We record the full trajectory (screens, actions, and final state), and human annotators

| Model | 0 (error) | 0.5 (partial) | 1 (success) | Overall |
|---|---|---|---|---|
| MobileRL-9B w/o ADAGRPO | 44.76% | 3.92% | 51.32% | 53.28% |
| MobileRL-9B | 42.25% | 3.67% | 54.09% | 55.92% |

Table 10: Results on the out-of-domain public-use apps evaluation suite (106 tasks across 21 apps). We report the empirical proportions of trajectories receiving scores 0, 0.5, and 1, and the overall average score. The full MobileRL-9B model with ADAGRPO achieves a higher average score and a larger fraction of fully successful trajectories, while reducing the fraction of complete failures.

assign a discrete score in $\{0, 0.5, 1\}$, where $0$ indicates failure (the final state does not satisfy the instruction), $0.5$ indicates partial success or unacceptable redundant operations, and $1$ indicates success without unacceptable redundancy. We report the average score over all 106 tasks, along with the empirical proportions of each score.

**Models and results.** We evaluate two agents in this out-of-domain setting:

1. **MobileRL-9B w/o ADAGRPO**: the agent trained with our standard pipeline but *without* the proposed difficulty-adaptive GRPO (ADAGRPO) reinforcement learning.

2. **MobileRL-9B**: the full agent with reinforcement learning.

Table 10 summarizes the proportions of each score category and the overall average score.

On this out-of-domain suite, MobileRL-9B with ADAGRPO improves the overall average score from 53.28% to 55.92% (+2.7 points). Since these apps and tasks are not used during RL, the results indicate that the gains from ADAGRPO are not limited to the in-domain benchmarks.

### E.2 DIFFICULTY-AWARE EVALUATION ON ANDROIDWORLD

#### E.2.1 ANDROIDWORLD ANNOTATIONS AND BASELINE DIFFICULTY

AndroidWorld provides public metadata for each evaluation task, including (i) a human-annotated difficulty label `difficulty_raw` (`easy`/`medium`/`hard`), (ii) an estimated optimal number of steps $S$ (shortest-path length), and (iii) semantic tags describing interaction and reasoning types (e.g., `multi_app`, `complex_ui_understanding`, `screen_reading`, `math_counting`, `memorization`, `information_retrieval`, `transcription`, `data_entry`).

All analyses below rely solely on this public metadata and deterministic rules over task templates; we do not introduce additional human annotation.

#### E.2.2 CONSTRUCTION OF ADDITIONAL DIFFICULTY DIMENSIONS

To disentangle horizon length from semantic/cognitive difficulty, we derive several complementary difficulty views based on AndroidWorld.

**Semantic / reasoning-related features** From the AndroidWorld tags and the task template text, we derive three binary features for each task. We set `is_temporal_reasoning=1` if the lowercased template contains temporal keywords (e.g., *week*, *today*, *tomorrow*, *date*, *time*); we set `is_aggregation_reasoning=1` if it contains aggregation cues (e.g., *how many*, *total*, *sum*, *duration*, *distance*); and we set `is_multi_goal=1` if it explicitly describes a multi-stage goal (e.g., contains *and then* or multiple clauses with *then*).

These features complement the AndroidWorld semantic tags by explicitly capturing different kinds of reasoning and coordination requirements: temporal reasoning, numerical aggregation, low-level system control, filesystem operations, and multi-stage goals.

**Semantic complexity score and buckets (S1/S2/S3)** We define a semantic complexity score by combining (i) reasoning-related tags $R = \{$`math_counting`, `memorization`, `information_retrieval`, `transcription`$\}$,

(ii) perception/UI-related tags $P = \{\texttt{complex\_ui\_understanding}, \texttt{screen\_reading}\}$, (iii) the presence of $\texttt{multi\_app}$, and (iv) the three binary features above.

Let $\texttt{tags\_set}$ be the set of AndroidWorld tags for a task. We compute:

$$\text{reason\_score} = |\texttt{tags\_set} \cap R|,$$
$$\text{perception\_score} = |\texttt{tags\_set} \cap P|,$$
$$\text{multi\_app\_flag} = \mathbb{I}[\texttt{multi\_app} \in \texttt{tags\_set}].$$

The semantic complexity score is then

$$
\begin{aligned}
\text{semantic\_score} = {} & \text{reason\_score} + \text{perception\_score} + \text{multi\_app\_flag} \\
& + \texttt{is\_temporal\_reasoning} + \texttt{is\_aggregation\_reasoning} \\
& + \texttt{is\_multi\_goal}.
\end{aligned}
$$

Weights are chosen to give higher importance to reasoning, multi-app composition, and multi-goal structure, while still accounting for perceptual/UI complexity and operational difficulty.

We sort tasks by semantic_score and split them into three equal-sized buckets: S1 (lowest tertile), S2 (middle tertile), and S3 (highest tertile).

**Combined difficulty score and buckets (D1/D2/D3)**   To obtain a unified one-dimensional notion of difficulty, we combine horizon and semantic complexity. Let $S$ be the optimal step count for a task, and define

$$\text{norm\_steps} = \frac{S - S_{\min}}{S_{\max} - S_{\min}},$$

the normalized step count in $[0, 1]$ over all tasks. The combined difficulty score is

$$\text{combined\_difficulty\_score} = 0.5 \cdot \text{norm\_steps} + 1.0 \cdot \text{semantic\_score}.$$

Here semantic complexity receives a higher weight (1.0) than horizon (0.5) to avoid simply treating long paths as intrinsically more difficult.

We sort tasks by combined_difficulty_score and again split them into three equal-sized buckets: D1 (low), D2 (medium), and D3 (high) combined difficulty.

### E.2.3 Evaluation Protocol and pass@k Computation

For each AndroidWorld task, we evaluate both MobileRL w/o AdaGRPO and MobileRL.

For a given task with $c$ successful rollouts out of $n$ total, we compute *pass@k* for $k \in \{1, 2, 4, 8\}$ using the standard unbiased estimator:

$$\text{pass@k} = 1 - \frac{\binom{n-c}{k}}{\binom{n}{k}},$$

with $k$ truncated to at most $n$. For each difficulty category (e.g., $\texttt{easy}$ vs. $\texttt{hard}$, H1 vs. H3, S1 vs. S3), we report the average pass@k across all tasks in that category for both the MobileRL w/o AdaGRPO and MobileRL models.

Here we provide some examples of tasks and their difficulty ratings in Table 11. Such ratings allow us to select long but simple tasks, as well as short tasks that require multiple types of understanding.

### E.2.4 Results by Difficulty

We summarize pass@k performance under all difficulty definitions. Full numerical results are provided in Tables 12–17. Below we highlight the main trends.

Across AndroidWorld's original difficulty labels (Table 12), the MobileRL model achieves consistent improvements over the MobileRL w/o AdaGRPO baseline at all difficulty levels, with particularly notable gains on $\texttt{medium}$ and $\texttt{hard}$ tasks. This shows that AdaGRPO is beneficial not only for easy, well-structured tasks but also improves the agent's robustness on challenging interactions.

Table 11: Examples of tasks and their human-annotated difficulty labels and structural properties. These annotations allow us to select long but simple tasks as well as short tasks that require multiple types of understanding.

| Task | Human-annotation Difficulty | Horizon Label | Semantic Label | Combined Label |
|------|------|------|------|------|
| What activities did I do {date} in the Open-Tracks app? Answer with the category only. If there are multiple categories, format your answer in a comma separated list. | hard | short | S3 | D3 |
| Create a playlist in Retro Music titled "{playlist_name}" with the following songs, in order: {names}. Then export the playlist to the Downloads directory on the device. | hard | medium | S2 | D3 |
| Add the following recipes into the Broccoli app: {recipes}. | medium | long | S1 | D1 |

Table 12: Results grouped by AndroidWorld's original difficulty label `difficulty`.

| Difficulty | SFT@1 | RL@1 | SFT@2 | RL@2 | SFT@4 | RL@4 | SFT@8 | RL@8 |
|------|------|------|------|------|------|------|------|------|
| easy | 0.773 | 0.867 | 0.795 | 0.878 | 0.811 | 0.889 | 0.823 | 0.908 |
| medium | 0.499 | 0.614 | 0.536 | 0.637 | 0.572 | 0.668 | 0.606 | 0.706 |
| hard | 0.208 | 0.247 | 0.230 | 0.289 | 0.248 | 0.313 | 0.260 | 0.316 |

Using our proposed semantic complexity buckets (Table 13), AdaGRPO improves performance across all three levels and substantially narrows the gap between low- and high-complexity tasks (e.g., strong relative gains on S3). This suggests that RL primarily strengthens reasoning, multi-step coordination, and UI-understanding capabilities.

The combined difficulty buckets (Table 14), which integrate horizon and semantic factors, show a similar pattern: while easy tasks (D1) benefit only slightly, the largest relative improvements occur in the highest combined difficulty bucket (D3). This indicates that AdaGRPO specifically boosts tasks requiring both long-horizon interaction and semantic reasoning.

Finally, the binary semantic features (Tables 15–17) further confirm this trend. AdaGRPO improves performance across all feature-defined splits (temporal reasoning, aggregation, and multi-goal structure). In particular, multi-goal tasks—representing some of the most structurally complex interactions—see substantially larger relative gains despite lower absolute performance.

Overall, across every difficulty definition, AdaGRPO consistently improves model performance, with the largest relative gains occurring in high-complexity settings. These results support our claim that AdaGRPO training meaningfully increases the agent's ability to handle difficult, long-horizon, and reasoning-intensive AndroidWorld tasks.

### E.3 INPUT MODALITY OF BASELINES

Some baselines operate under different state representations (e.g., screenshot-only vs. screenshot+XML). When reporting baseline results, we follow a consistent principle: (i) prefer the best officially reported score when available; otherwise, (ii) evaluate using the model's recommended phone-use format; and otherwise (iii) evaluate within our pipeline and explicitly state the input format.

In our comparisons, models such as GPT-4o are evaluated with screenshot-only input, while some models without an official phone-operation template are evaluated using our default state representation (screenshot+XML). For GUI-specialized agents with prescribed templates (e.g., UI-Tars), we follow their official operation templates to avoid artificially lowering performance.

Table 13: Results grouped by semantic complexity bucket `semantic_bucket` (S1/S2/S3).

| Semantic | SFT@1 | RL@1 | SFT@2 | RL@2 | SFT@4 | RL@4 | SFT@8 | RL@8 |
|---|---|---|---|---|---|---|---|---|
| S1 | 0.677 | 0.758 | 0.704 | 0.773 | 0.729 | 0.799 | 0.749 | 0.840 |
| S2 | 0.583 | 0.653 | 0.604 | 0.698 | 0.627 | 0.728 | 0.648 | 0.744 |
| S3 | 0.487 | 0.615 | 0.518 | 0.618 | 0.538 | 0.618 | 0.552 | 0.618 |

Table 14: Results grouped by combined difficulty bucket `combined_difficulty_bucket` (D1/D2/D3).

| Combined | SFT@1 | RL@1 | SFT@2 | RL@2 | SFT@4 | RL@4 | SFT@8 | RL@8 |
|---|---|---|---|---|---|---|---|---|
| D1 | 0.791 | 0.862 | 0.818 | 0.874 | 0.845 | 0.890 | 0.862 | 0.913 |
| D2 | 0.523 | 0.605 | 0.543 | 0.630 | 0.560 | 0.656 | 0.580 | 0.697 |
| D3 | 0.468 | 0.592 | 0.502 | 0.615 | 0.527 | 0.629 | 0.546 | 0.632 |

### E.4 VISION-ONLY SETTING AND ROBUSTNESS TO MISSING XML

XML metadata may be incomplete for some real-world apps, and some clickable regions may not be covered by the XML hierarchy. We therefore assess two settings on AndroidWorld: (i) evaluating the same trained MobileRL-9B checkpoint with screenshot-only input (i.e., removing XML at test time), and (ii) re-running the full pipeline in a pure vision-only mode where state representations and action grounding are based on screenshots only.

Evaluating MobileRL-9B under the extreme XML-unavailable condition yields a non-trivial score of 24.6% on AndroidWorld, despite the mismatch between the training and testing formats.

We also re-trained the full pipeline in the vision-only mode (using GLM-4.1V-9B-Base as the backbone) and obtained the following results on AndroidWorld:

| Stage | Screenshot-only | Screenshot+XML |
|---|---|---|
| + Reasoning-Free SFT | 47.5 | 48.1 |
| + Reasoning SFT | 62.6 | 66.2 |
| + AdaGRPO | 79.3 | 80.2 |

These results suggest that, with the current model capacity and training recipe, a vision-only training pipeline can approach the performance of screenshot+XML, while we expect further improvements with stronger visual grounding.

### E.5 IMPACT OF THE ANDROIDLAB REWARD MODEL.

To better understand the bias introduced by the AndroidLab reward model, we first randomly sample 300 trajectories from different historical checkpoints and manually annotate their labels. The resulting confusion matrix is $TP = 100$, $FP = 34$, $TN = 155$, and $FN = 11$, corresponding to an overall accuracy of $0.85$, precision of $0.74$, recall of $0.91$, and F1 score of $0.82$. These statistics indicate that the dominant error type is false positives, where incorrect trajectories are mistakenly classified as successful. Manual inspection suggests that the reward model sometimes over-relies on the textual `think` and `action` descriptions while overlooking whether the corresponding GUI operations have actually taken effect on the screen.

To further quantify the impact of this $86\%$-accuracy reward model on RL, we perform an additional experiment. We take the original AndroidLab test set (which consists entirely of tasks with verifiable rewards), randomly split it into two equal subsets, and treat one subset as a new training set and the other as a new test set. We then re-run the RL algorithm under two conditions: (i) using the verifiable reward as the trajectory-level reward during training, and (ii) using the AndroidLab reward model as the trajectory-level reward, while in both cases evaluating with the *verifiable* reward on the test subset, as shown in Figure 15. Starting from the same initial verified success rate of $44.4$, training with verifiable rewards improves the test verified reward to $63.7$, whereas training with the reward

Table 15: Results grouped by temporal reasoning indicator `is_temporal_reasoning`.

| is_temporal _reasoning | SFT@1 | RL@1 | SFT@2 | RL@2 | SFT@4 | RL@4 | SFT@8 | RL@8 |
|---|---|---|---|---|---|---|---|---|
| 0 | 0.616 | 0.705 | 0.642 | 0.721 | 0.664 | 0.743 | 0.683 | 0.769 |
| 1 | 0.543 | 0.671 | 0.578 | 0.704 | 0.617 | 0.742 | 0.651 | 0.780 |

Table 16: Results grouped by aggregation/counting indicator `is_aggregation_reasoning`.

| is_aggregation _reasoning | SFT@1 | RL@1 | SFT@2 | RL@2 | SFT@4 | RL@4 | SFT@8 | RL@8 |
|---|---|---|---|---|---|---|---|---|
| 0 | 0.618 | 0.701 | 0.644 | 0.717 | 0.665 | 0.738 | 0.684 | 0.765 |
| 1 | 0.463 | 0.661 | 0.500 | 0.700 | 0.543 | 0.739 | 0.580 | 0.778 |

model improves it to 59.1. This result indicates that, under our training configuration, the errors of the AndroidLab reward model do not get excessively amplified by the RL process.

## F    EXTENDED BACKGROUND ON PRIOR APPROACHES

**Length-related penalties and controlling reasoning length.**    Recent RLVR work for LLMs increasingly treats chain-of-thought (CoT) length as an explicit optimization dimension. LASER proposes step-wise length-based reward shaping and adaptive variants (LASER-D/DE) that tune the effective length budget according to task difficulty, achieving better accuracy–efficiency trade-offs on math benchmarks (Liu et al., 2025). LACONIC instead augments the RL objective with a length cost applied only to tokens beyond a target budget, enforcing token limits during training while preserving task reward as the main signal (Anonymous, 2025). Thinking Fast and Right dynamically adjusts the weight of a length penalty based on current model accuracy, increasing the penalty when the model is already accurate and relaxing it when accuracy drops, thereby stabilizing the trade-off between correctness and length (Su & Cardie, 2025). All of these can be interpreted as practical instances of potential-based reward shaping (Ng et al., 1999) specialized to sequence length.

Our SPA reward extends length-based shaping ideas from text-only RLVR for LLMs to embodied GUI agents, but now operates on trajectory length rather than token count. Instead of relying on raw token length or judge-defined conciseness, SPA uses the empirically observed shortest successful trajectory length for each mobile task as a data-driven baseline and rescales the terminal reward of each successful rollout according to its relative path length. In doing so, SPA simultaneously constrains correct trajectories from degenerating into unnecessarily long, looping explorations, while avoiding the opposite failure mode of naively rewarding early termination that leads to incorrect outcomes: only successful trajectories are length-shaped, and their rewards are modulated relative to the best-known path for the same task. This yields a group-relative, trajectory-level shaping that is compatible with binary verifiable rewards in GRPO, directly reflecting interaction efficiency in the mobile GUI environment.

**Curriculum learning.**    In classical reinforcement learning, curriculum learning studies how to order tasks or training samples so that agents acquire complex skills gradually; Narvekar et al. (2020) provides a general framework over task generation, ordering, and stopping criteria, and surveys teacher–student and self-paced methods in Atari and continuous control domains. In the LLM era, curriculum ideas have been adapted to RL-based post-training: WebRL implements a *self-evolving curriculum* for web agents by turning failure trajectories into new navigation tasks and feeding them back into an online RL loop with a learned reward model, so that the task distribution adapts as the agent improves (Qi et al.). Sun et al. (2025) further couples RLVR with curriculum-like difficulty targeting via *Difficulty-targeted Online Data Selection*, which selects prompts whose current success rate is around $50\%$ and combines this with rollout replay to focus GRPO-style training on "on-the-margin" problems.

In our setting, we adopt a variant of curriculum learning through failure-centric curriculum filtering. Rather than only up-weighting medium-difficulty prompts, FCF explicitly tracks commands that

Table 17: Results grouped by multi-goal indicator `is_multi_goal`.

| is_multi _goal | SFT@1 | RL@1 | SFT@2 | RL@2 | SFT@4 | RL@4 | SFT@8 | RL@8 |
|---|---|---|---|---|---|---|---|---|
| 0 | 0.625 | 0.707 | 0.652 | 0.723 | 0.674 | 0.739 | 0.691 | 0.762 |
| 1 | 0.042 | 0.317 | 0.075 | 0.407 | 0.120 | 0.484 | 0.158 | 0.500 |

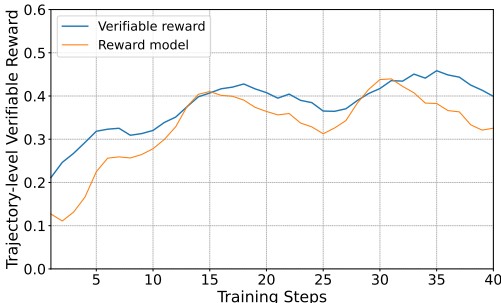

Figure 15: **AndroidLab RL training with verifiable vs. reward-model rewards.**Trajectory-level verifiable reward on the AndroidLab subset when training with (i) verifiable rewards and (ii) the AndroidLab reward model as the trajectory-level signal.

remain persistently unsolved in the mobile GUI environment and down-weights or temporarily cools them in the sampling distribution. This allows us, under strict interaction and compute budgets, to concentrate RL updates on a subset of tasks that are more likely to yield improvement, thereby increasing training efficiency while still maintaining exposure to harder tasks over time.

**Replay-based methods.**   Experience replay has long been a key mechanism to improve sample efficiency and stabilize learning in off-policy RL. Schaul et al. (2016) introduce *Prioritized Experience Replay* (PER), in which transitions are sampled from a replay buffer with probabilities proportional to their temporal-difference error, so that more "informative" experiences are reused more often; they show substantial gains in Deep Q-Networks on Atari compared to uniform replay. In large-scale RL for LLM-based device-control agents, DistRL employs an asynchronous distributed framework in which many workers interact with Android environments in parallel and send full trajectories to a centralized replay buffer; its A-RIDE algorithm uses *Distributed Prioritized Experience Replay (DPER)*, assigning each trajectory a priority that is a weighted combination of its average absolute TD error, average importance-sampling ratio, and average policy entropy, so that highly off-policy, high-error, and exploratory trajectories are replayed more frequently, with priorities periodically recomputed under the latest policy (Wang et al., 2024b). In LLM post-training, Trajectory Balance with Asynchrony (TBA) decouples exploration and learning: a swarm of searcher nodes continuously populates a global replay buffer with off-policy trajectories, while a trainer node samples from this buffer based on reward or recency to optimize a trajectory-balance objective for LLMs (Bartoldson et al., 2025).

Motivated by the high sampling cost and heavy-tailed difficulty of mobile GUI tasks, we introduce AdaPR as a prioritized replay scheme tailored to our setting. Instead of generic replay keyed by TD-error or recency, we maintain a buffer of successful, high-advantage trajectories, and assign higher sampling probability to rare but informative successes while still mixing in fresh on-policy rollouts. In this way, AdaPR makes the most of expensive environment interactions, maximally reuses high-advantage behavior, and stabilizes GRPO training. Moreover, it is tightly integrated with our SPA reward shaping and FCF curriculum modules, rather than being an isolated system-level optimization.

## G    THE USE OF LARGE LANGUAGE MODELS

We claim that during the whole process of this work, Large Language Models(LLM) are used only to polish our manuscript for better readability. We first draft a paper version without any help from LLM. Upon drafting this version, we applied LLM to check for typos, grammar errors, and possible ambiguities in the writing expression.

Notably, no LLM usage is applied in the research methodology or experimental design of this work. All ideas in algorithm design and experiment settings come from the authors of the paper. All data analysis and experiments are actually carried out by the authors of the paper.

## H    LIMITATION

**Reward Model**    We train a reward model to gain reward signals during the training on the Android-Lab environment. The training data is obtained from strong proprietary VLMs through majority voting, but it is noteworthy that bias still exists in those data. Therefore, the reward model is inevitably born slightly biased from a perfect judge on agentic tasks, which brings about unsatisfying results during online RL in some cases. In particular, while MOBILERL-7B and all ablation studies are carried out with the existence of the reward model, the best model, MOBILERL-9B, is gained from online RL purely on the AndroidWorld environment. In observation of this, we conclude that a general and unbiased verification model is of significant importance and necessity, which is proposed as one aspect of our future work.

**Difficulties with Public-Use Apps**    We evaluate out-of-domain generalization on public-use apps via a human-evaluated suite (Appendix E.1). However, training and evaluating on public-use apps at scale remains challenging due to the lack of a reproducible and efficient protocol in the open domain. Existing approaches typically require human execution of tasks and manual assessment of outcomes, which are costly and difficult to scale.

Moreover, public-use apps often introduce practical challenges. Many require user login, cannot be run reliably on virtual machines, or lack mechanisms to restore application states, making large-scale online rollouts particularly challenging from an engineering perspective. We plan to explore solutions for more scalable training and evaluation on public-use apps in future work.

**Limitation of prediction based on XML**    We observed that certain applications on the Android platform, especially those with large popularity and of comprehensive functions(e.g. TikTok, etc.), lack sufficient information about all interactable elements in their XML hierarchy data. To successfully perform complex tasks on these Apps, the current format we chose in model prediction, which is based on bounding box information contained in XML data, needs adaptation toward the general format built on solely visual information. This adaptation is, in the end, inevitable while challenging, so we leave it for our future work to obtain a generalized vision model with no need for XML data.

**Training Time and Resource Consumption**    Another limitation of our work lies in the substantial resource demands of Android virtual devices. Each device requires more than 5 GB of memory and over 30 GB of storage, which restricts the number of rollouts that can be run concurrently. To address this, we employed four CPU machines with 1 TB of memory each, dedicated solely to rollouts. Although we implemented a distributed framework that supports multi-machine concurrent sampling, the system can only stably sustain up to 256 parallel rollouts, with each step taking more than 10 minutes.

This constraint severely limits further scaling of training steps, as training 100 steps already takes more than 25 hours. In addition, large-scale image inference and network transmission introduce significant overhead, further increasing the computational burden.

