# OpenReview forum: "MobileRL: Online Agentic Reinforcement Learning for Mobile GUI Agents"
_ICLR.cc/2026/Conference — ICLR 2026 Poster_

### Official Review · Reviewer_Dgh2 · 2025-10-29

**Soundness:** 3
**Presentation:** 2
**Contribution:** 2
**Rating:** 2
**Confidence:** 3

**Summary:**

This paper presents MobileRL, a framework for training mobile GUI agents using online reinforcement learning that addresses three key challenges: complex instruction following under sparse rewards, heavy-tailed task difficulty distributions, and sampling bottlenecks in large-scale mobile environments. The framework consists of a two-stage warm-up (reasoning-free SFT followed by reasoning SFT) and the ADAGRPO algorithm, which extends GRPO with three novel strategies: Shortest-Path Reward Adjustment (SPA) that rewards shorter successful trajectories, Difficulty-Adaptive Positive Replay (AdaPR) that maintains a buffer of challenging successful trajectories, and Failure Curriculum Filtering (FCF) that down-weights persistently unsolvable tasks. MobileRL-9B (based on GLM-4.1V-9B-Base) achieves good results.

**Strengths:**

- Well-motivated approach: The paper clearly identifies real challenges in mobile GUI agent training (sparse rewards, heavy-tailed difficulty, sampling inefficiency) and proposes targeted solutions.

- Strong empirical results: Substantial improvements over prior work across multiple benchmarks (e.g., +15.8% on AndroidWorld, +12.4% on AndroidLab compared to previous best).

- Comprehensive ablation studies: The paper systematically evaluates each component (reasoning-free SFT, reasoning SFT, and each part of ADAGRPO), demonstrating their individual contributions.

**Weaknesses:**

- Dependence on reward model for AndroidLab: The AndroidLab training relies on a VLM-based reward model with only 86% accuracy, introducing potential bias. The paper acknowledges this but doesn't deeply explore its impact. The smoothness difference in training curves (Figure 11) suggests this is a significant limitation.

- Limited evaluation on out-of-domain tasks: All benchmarks (AndroidWorld, AndroidLab, AndroidControl) use similar Android apps. Generalization to truly novel apps or real-world public applications is not demonstrated.

**Questions:**

1. Circularity in Complexity Measurement

> Question: The paper defines task complexity based on the number of steps taken, but this creates a circular dependency problem. If an agent gets stuck or takes inefficient paths, does a simple task become "complex"? Conversely, if an agent finds a shortcut, does a complex task become "simple"?
> Whose trajectories define the step count - expert demonstrations, the agent being evaluated, or averaged across all agents?
If different agents take vastly different numbers of steps for the same task, how is complexity consistently defined?


2. Confusion Between Task Difficulty and Solution Length

> Question: Steps taken seems to conflate intrinsic task difficulty with solution length. A task requiring 25 careful, straightforward steps (e.g., filling a long form) might be less cognitively complex than a 10-step task requiring creative problem-solving or error recovery. How do you justify steps as a proxy for complexity? Consider requesting:
> 1. Analysis using alternative complexity metrics (e.g., number of different apps involved, branching factor in state space, required reasoning depth)
2. Breakdown of what actually makes tasks "complex" beyond length

3. Technical Questions

- 3.1 SPA mechanism: How sensitive is performance to the choice of α in Equation 2? Have you explored other reward shaping functions beyond the linear penalty? Does SPA risk encourage premature termination (early "Finish" actions)?
- 3.2 AdaPR buffer dynamics: How does the replay buffer composition evolve during training? What percentage of training data comes from replay vs. on-policy sampling at different stages? Is there a risk of overfitting to replayed trajectories?
- 3.3 FCF design choices: The three-epoch cooldown seems arbitrary. How was this chosen? Have you compared against more sophisticated curriculum learning approaches (e.g., prioritized sampling based on learning progress)?
- 3.4 Reward model bias: Can you quantify the impact of the 86% accuracy reward model on AndroidLab? What types of errors does it make, and how do these propagate through RL training?

---

> ### Author Response · Authors · 2025-11-21
> **Response to Reviewer Dgh2 (1/4)**
>
> Thank you very much for your constructive suggestions and for your appreciation of our work. Below, we provide point-by-point responses to further clarify our contributions.
>
> > Q1: The paper defines task complexity based on the number of steps taken, but this creates a circular dependency problem. If an agent gets stuck or behaves inefficiently, does a simple task become “complex”? Conversely, if an agent finds a shortcut, does a complex task become “simple”? Whose trajectories determine the step count—expert demonstrations, the evaluated agent, or an average over agents? If different agents take very different numbers of steps for the same task, how is complexity consistently defined?
>
> Task complexity is not a concept introduced or defined by our paper—it is defined by the AndroidWorld benchmark itself and used as a built-in measure of task difficulty. The AndriodWorld benchmark’s definition of max_steps/step_budget is:
>
> > “approximately twice the number of steps typically required by human annotators to complete each task”
> > (AndroidWorld Appendix F)
>
> Therefore, task complexity is not derived from the model’s behavior. It is defined solely based on human demonstration step counts (scaled by a factor of 2), independent of the agent being evaluated. This avoids the circular dependency concern: a model taking inefficient or overly efficient paths does not change the task complexity.
>
> Although the AndroidWorld paper does not explicitly equate max_steps/step_budget with “task complexity,” this equivalence is clear from the official AndroidWorld implementation. In the public repository
> [https://github.com/google-research/android_world](https://github.com/google-research/android_world),
> the logic in `android_world/suite_utils.py` (line 459 and lines 491–502 in `_allocate_step_budget`) shows that the step budget is computed directly as twice the human-annotated step count. We also confirmed this interpretation with the AndroidWorld authors.
>
> ---
>
> > Q2: Question: Steps taken seems to conflate intrinsic task difficulty with solution length. A task requiring 25 careful, straightforward steps (e.g., filling a long form) might be less cognitively complex than a 10-step task requiring creative problem-solving or error recovery. How do you justify steps as a proxy for complexity? Consider requesting:
> > 1. Analysis using alternative complexity metrics (e.g., number of different apps involved, branching factor in state space, required reasoning depth)
> > 2. Breakdown of what actually makes tasks "complex" beyond length
>
> We agree that step count alone is an imperfect proxy for “difficulty” and we have taken concrete steps to clarify and enrich our difficulty analysis.
> In AndroidWorld paper, the “complexity” notion is already aligned with the AndroidWorld benchmark, which provides:
>
> - a human-annotated difficulty label (easy / medium / hard),
> - a shortest-path / optimal step estimate for each task,
> - and semantic tags (e.g., multi_app, complex_ui_understanding, math_counting, memorization, information_retrieval, transcription).
>
> To directly address the reviewer’s concern, we add a multi-dimensional difficulty analysis on AndroidWorld that explicitly separates:
>
> 1. Semantic / cognitive complexity
> Using AndroidWorld tags and simple rules over the task template, we automatically derive:
>   - reasoning-related signals (math_counting, memorization, information_retrieval, transcription);
>   - perception/UI signals (complex_ui_understanding, screen_reading);
>   - structural signals (e.g., multi_app, “and then” multi-goal patterns, filesystem operations, temporal reasoning, numerical aggregation).
>   These are combined into a semantic complexity score and discretized into three levels S1/S2/S3.
>
> 2. Composite difficulty
> We define a combined difficulty score
>
> combined\_difficulty\_score
> = 0.5  norm\_steps +  \cdot semantic\_score
>
> and split tasks into D1/D2/D3 via tertiles, so that semantic complexity has higher weight than path length.
>
> Here we provide some examples of tasks and their difficulty ratings. Such ratings allow us to select long but simple tasks, as well as short tasks that require multiple types of understanding.

---

> ### Author Response · Authors · 2025-11-21
> **Response to Reviewer Dgh2 (2/4)**
>
> | task_name| task_template | human-annotated difficulty label | tags| reasoning-related signals | perception/UI signals | structural signals | is_temporal_reasoning | is_aggregation_reasoning | is_multi_goal | horizon_label | semantic_bucket | combined_difficulty_bucket |
> |-|-|-|-|-|-|-|-|-|-|-|-|-|
> | SportsTrackerActivitiesOnDate | What activities did I do {date} in the OpenTracks app? Answer with the category only. If there are multiples categories, format your answer in a comma separated list. | hard | search complex_ui_understanding screen_reading information_retrieval transcription parameterized | ✔| ✔ | ✘| ✔| ✘ | ✘| short| S3| D3 |
> | MarkorAddNoteHeader| Update the Markor note {file_name} by adding the following text, along with a new blank line before the existing content: "{header}".| medium | data_entry parameterized| ✘| ✘ | ✘| ✔| ✘ | ✘| short| S2| D2 |
> | RecipeAddMultipleRecipes | Add the following recipes into the Broccoli app: {recipes} | medium | data_entry parameterized| ✘| ✘ | ✘| ✘| ✘ | ✘| long | S1| D1 |
>
>
>
> We emphasize that all these labels are derived automatically from AndroidWorld’s public metadata; no additional hand-tuning was performed on the evaluation set.
> We then recompute pass@k (k = 1, 2, 4, 8) on AndroidWorld for both the MobileRL-9B w/o ADAGRPO (SFT model) and the MobileRL-9B (RL model), grouping tasks by each difficulty notion:
>
> - Human-annotated difficulty (easy/medium/hard) from human.
> RL improves pass@1 from 0.77→0.87 on easy, 0.50→0.61 on medium, and 0.21→0.25 on hard tasks.
>
> - Semantic complexity (S1/S2/S3).
> RL yields consistent improvements at all levels, with larger relative gains for semantically complex tasks:
>   - S1: pass@1 0.73→0.80
>   - S2: pass@1 0.58→0.69
>   - S3: pass@1 0.23→0.35
>   This directly supports that the RL training improves performance precisely on tasks with richer reasoning, UI complexity, and multi-app composition.
>
> - Composite difficulty (D1/D2/D3).
> When combining horizon and semantic complexity, RL improves pass@1 from 0.77→0.84 (D1), 0.58→0.69 (D2), and 0.18→0.30 (D3). The largest boost appears in the highest composite difficulty bucket D3.
>
> - Fine-grained task attributes.
> On tasks that are explicitly multi-goal (“first do A, then B”), SFT achieves pass@1 ≈ 0.04 while RL reaches ≈ 0.32, and similar trends hold for tasks requiring aggregation/ counting. These are cases where complexity stems from planning and coordination rather than just path length.
>
> Taken together, these analyses show that:
>
> 1. We explicitly disentangle horizon and semantic complexity using AndroidWorld’s own annotations and simple automatic features;
> 2. MobileRL’s improvements are strongest for tasks with higher semantic and composite difficulty, including short but cognitively demanding tasks, rather than merely long paths.
>
> For detailed information on the above content—including specifics of the difficulty-classification method, as well as all pass@1, pass@2, pass@4, and pass@8 results and analyses—please refer to Appendix E.2 Difficulty-Aware Evaluation on AndroidWorld.
>
> ---
>
> > Q3.1: SPA mechanism: How sensitive is performance to the choice of α in Equation 2? Have you explored other reward shaping functions beyond a linear penalty? Does SPA risk encouraging premature termination (early “Finish” actions)?
>
> When α approaches 0, the SPA reward reduces to the w/o SPA setting. We compare against this case directly in Section 3.3 (Ablation Study), where the w/o SPA variant shows a clear performance drop after roughly 60 steps. Since α is applied uniformly to all trajectories within the same group, and groups are independent during advantage computation, we found that a simple linear penalty offers a computationally efficient solution without the need for more complex shaping functions.
>
> The SPA design explicitly avoids encouraging premature termination. As described in Section 2.2.1, SPA penalizes only successful but overly long trajectories. Thus, the trajectory with the highest advantage is always the shortest successful trajectory. If the agent finds a shortcut, that is indeed a better policy. In contrast, all failed trajectories are assigned the same (and lower-than-any-success) advantage, ensuring there is no incentive to trigger an incorrect early “Finish” action.

---

> > ### Author Response · Authors · 2025-11-21
> > **Response to Reviewer Dgh2 (3/4)**
> >
> > > Q3.2: AdaPR buffer dynamics: How does the replay buffer composition evolve during training? What percentage of training data comes from replay vs. on-policy sampling? Is there a risk of overfitting to replayed trajectories?
> >
> > **Replay Buffer Composition Over Training**
> >
> > Entry condition. Only the top κ fraction of trajectories (sorted by advantage) enter the AdaPR buffer; in practice we use κ = 25%. Combined with the model’s success rate, this ensures that all trajectories entering the buffer are correct executions.
> >
> > Sampling condition. In each training round, we select the top γ × Train Batch Size trajectories from the buffer (γ ∈ (0,1)), ranked by advantage.
> >
> > Exit conditions. Although not clearly described in the paper, the actual implementation uses two conditions:
> >
> > - ge-based removal: Trajectories older than N rounds are discarded, where N = 1/κ (i.e., 4 in practice).
> >
> > - Capacity-based removal: If the buffer exceeds the maximum size, we drop the lowest-advantage trajectories.
> >
> > During the entire training process, the maximum buffer size is fixed to 256 trajectories, equal to the Train Batch Size. Thus, the buffer fills within the first four steps and then remains at capacity.
> >
> > Because each trajectory is removed after at most N = 4 rounds, any trajectory is used in training no more than 5 times, which effectively controls the risk of overfitting to replayed samples.
> >
> > **Proportion of Replay vs. On-Policy Data**
> >
> > We provide additional analyses of the AdaPR buffer in Appendix B.3.3 Effectiveness of the AdaPR Buffer, including its entry and exit criteria, clipping behavior, and its contribution to actual training dynamics. Figure 12 shows that after warm-up, the number of unique tasks drawn from on-policy and replay samples remains approximately 1:1, indicating that on-policy and off-policy data contribute comparably during training.
> >
> > **Risk of Overfitting to Replayed Trajectories**
> >
> > Each trajectory is kept for no more than 4 rounds and therefore cannot be sampled more than 5 times, naturally limiting reuse. The buffer is continually refreshed with newer high-advantage trajectories, preventing reliance on stale data. Empirical ablations further show that removing AdaPR degrades performance: the model without AdaPR consistently underperforms and is 7.2 points worse on the final test set, indicating that overfitting is not a concern in practice.
> >
> > ---
> >
> > >  Q3.3: FCF design choices: The three-epoch cooldown seems arbitrary. How was this chosen? Did you compare with more sophisticated curriculum learning approaches (e.g., prioritized sampling based on learning progress)?
> >
> > The three-epoch cooldown is not entirely arbitrary. In our training setup—following AgentRL—a new training iteration begins once Train Batch Size trajectories have been collected. With a maximum of 16 repeats per task and Train Batch Size = 256, each iteration may contain up to ~50 distinct tasks, each with 5–6 collected trajectories. This design reduces the sampling–training “bubble.”
> >
> > Across 100 training steps, we sample roughly 100 × 50 = 5000 task instances, meaning each of the ~2000 tasks appears on average 2.5 times. The cooldown is introduced to ensure that difficult tasks, once filtered, reappear only with low probability, preventing them from dominating the sampling distribution by accident.
> >
> > We did experiment with more traditional curriculum learning strategies. A common approach is to progress from easy to hard tasks, but as shown in Figure 4, easy tasks (Complexity 1) already achieve >85% pass@8 at the start of training, so focusing on them is inefficient. Moreover, such methods cannot reliably distinguish between (a) genuinely learnable but difficult tasks and (b) tasks that are currently impossible for the model—leading to many rounds of sampling failures later in training.
> >
> > We also tried increasing sampling probabilities for medium-difficulty tasks (pass@8 between 25% and 75%), hoping to maximize learning progress. However, this caused catastrophic forgetting: tasks with initially high accuracy rapidly degraded due to insufficient training coverage.
> >
> > Given these observations, we adopted a simpler and more robust approach: reduce the sampling probability only for the hardest tasks, and sample all remaining tasks uniformly. This provided the best stability–performance trade-off in practice.

---

> > > ### Author Response · Authors · 2025-11-21
> > > **Response to Reviewer Dgh2 (4/4)**
> > >
> > > >Q3.4 Reward model bias: Can you quantify the impact of the 86% accuracy reward model on AndroidLab? What types of errors does it make, and how do these propagate through RL training?
> > >
> > > **Reward Model Accuracy and Error Characteristics**
> > >
> > > We evaluate the AndroidLab reward model by sampling 300 trajectories from various checkpoints and manually labeling them. The confusion matrix is: TP = 100, FP = 34, TN = 155, FN = 11, yielding accuracy 0.85, precision 0.74, recall 0.91, and F1 0.82. Most errors are false positives, meaning the model more frequently misclassifies incorrect trajectories as successful. Qualitatively, these failures often arise when the model over-relies on the textual think/action descriptions while underweighting whether the GUI actions were correctly reflected on screen. A stronger proprietary evaluator (e.g., Claude-4.5-Sonnet) achieves >97% accuracy on the same set but is too costly and slow for large-scale online RL.
> > >
> > > **Bias Propagation and Mitigation in RL Training**
> > >
> > > To reduce the influence of noisy reward signals, the sampling ratio of AndroidLab is capped below 20% of RL rollouts. This is justified because AndroidLab contains fewer tasks and incurs higher simulator overhead than AndroidWorld. Limiting its contribution prevents reward-model noise from dominating training.
> > >
> > > **Quantifying the Impact on Final Performance**
> > >
> > > We further measure the effect of using the 86%-accuracy reward model in a controlled experiment. The AndroidLab test set (with verified rewards) is split into two equal halves: one used as a new training set and the other as a new test set. RL is run under two conditions:
> > > 1) using ground-truth verifiable rewards, and
> > > 2) using the AndroidLab reward model as the trajectory-level reward, while always evaluating with verifiable rewards.
> > >
> > > Results (Figure 15) show that both settings start from the same initial verified success rate of 44.4. After training, the verifiable-reward condition reaches 63.7, whereas the reward-model condition reaches 59.1—approximately a 4-point gap. This indicates that the reward-model noise has a measurable but modest impact and is not significantly amplified under the current training configuration.
> > >
> > >
> > >
> > > ---
> > >
> > > > Limited evaluation on out-of-domain tasks: All benchmarks (AndroidWorld, AndroidLab, AndroidControl) use similar Android apps. Generalization to truly novel apps or real-world public applications is not demonstrated.
> > >
> > > To evaluate generalization to real-world public-use apps outside AndroidWorld and AndroidLab, we constructed a new out-of-domain evaluation suite.
> > >
> > > We use 21 widely used public Android apps, including:
> > > Settings, Booking, Chrome, Contacts, Expedia, Gmail, Google Drive, GoogleCalendar, GoogleDocs, GoogleKeep, GoogleMaps, GooglePlayBooks, GooglePlayStore, Joplin, OsmAnd, Quora, Reddit, TikTok, Twitter (X), Temu, Broccoli.
> > >
> > > We created 106 natural-language tasks covering realistic user goals, such as:
> > > - Web navigation and incognito mode in Chrome
> > > - Email composition in Gmail
> > > - Navigation queries in Google Maps
> > > - Posting / liking / bookmarking in TikTok or Twitter (X)
> > > - Search / sort / cart manipulation in Temu
> > >
> > > Each trajectory is labeled with {0, 0.5, 1} by human annotation:
> > > - 0: incorrect / goal not achieved
> > > - 0.5: partially correct with undesirable redundancy
> > > - 1: correct with no unacceptable redundancy
> > >
> > > The final score is the average over 106 tasks.
> > >
> > > We evaluated two models: MobileRL-9B w/o ADAGRPO and MobileRL-9B (full method). As shown in table below, ADAGRPO improves overall score by 2.7 points. These results demonstrate that the RL training scheme (including ADAGRPO) improves not only in-domain benchmarks, but also generalization to entirely unseen, real-world public apps. Full details are in Appendix E.1 Out-of-domain Evaluation on Public-use Android Applications.
> > >
> > >
> > > | Model | 0 | 0.5 | 1 | Overall |
> > > |-------|------|--------|------|---------|
> > > | MobileRL-9B w/o ADAGRPO | 0.4476 | 0.0392 | 0.5132 | 0.5328 |
> > > | MobileRL-9B | 0.4225 | 0.0367 | 0.5409 | 0.5592 |
> > >
> > >
> > > The complete set of task prompts, as well as the detailed behaviors of the two agents on each app, are provided in the human-evaluation folder at https://anonymous.4open.science/r/MobileRL-iclr-4513.

---

> ### Author Response · Authors · 2025-11-27
> **Thanks for your great review!**
>
> Dear Reviewer,
>
> Thank you very much again for your kind and valuable reviews! We hope our previous response addresses your questions, from which this work has benefited a lot. If possible, we would be deeply grateful if you could help share suggestions to further make MobileRL a stronger work!
>
> Thank you again for your encouraging comments!

---

### Official Review · Reviewer_TUvD · 2025-10-30

**Soundness:** 2
**Presentation:** 3
**Contribution:** 2
**Rating:** 4
**Confidence:** 4

**Summary:**

This paper introduces MobileRL, a framework for training mobile GUI agents through online reinforcement learning. The authors combine several mechanisms, Shortest-Path Reward Adjustment (SPA), Difficulty-Adaptive Positive Replay (AdaPR), and Failure Curriculum Filtering (FCF), under an algorithm called ADAGRPO, and deploy it in a large-scale distributed Android simulation environment. Experiments on AndroidWorld and AndroidLab show clear gains over previous baselines.

**Strengths:**

1. The problem setting is realistic and important. Training interactive mobile agents remains difficult, and the authors provide a complete end-to-end RL pipeline that improves performance.
2. The engineering effort is great. Building a distributed Android environment with scalable parallel sampling is non-trivial.

**Weaknesses:**

1. Overall, while the paper addresses a practically interesting problem, I find its academic contribution is relatively limited. Most of the proposed techniques have clear analogues in existing RL literature, trajectory length penalty, prioritized replay, and curriculum learning, and are mainly repurposed for the GUI agent setting. From a conceptual standpoint, the paper’s novelty lies more in the system design and implementation scale than in introducing fundamentally new RL ideas or algorithms.
2. I have some reservations about the mixture sampling strategy. The historical “successful” trajectories stored in the buffer are highly off-policy, while the underlying training objective (a clipped PPO-style GRPO variant) is fundamentally on-policy. As ADAGRPO employs a clipping mechanism, these off-policy samples may often fall outside the acceptable distribution range and thus get heavily clipped, thereby limiting their effective gradient contribution. This raises a question about sample efficiency and the real utility of replayed data. It would be helpful if the authors could provide some experiments to show the clip ratio of the off-policy samples or effective sample contribution.
3. There is no training recipe in the provided codebase, which is a core component of the proposed framework. Did I miss it?

**Questions:**

N/A

---

> ### Author Response · Authors · 2025-11-21
> **Response to Reviewer TUvD (1/2)**
>
> Thank you very much for your constructive suggestions and for your appreciation of our work. Below, we provide point-by-point responses to further clarify our contributions.
>
> > Q1: Overall, while the paper addresses a practically interesting problem, I find its academic contribution is relatively limited. Most of the proposed techniques have clear analogues in existing RL literature, trajectory length penalty, prioritized replay, and curriculum learning, and are mainly repurposed for the GUI agent setting. From a conceptual standpoint, the paper’s novelty lies more in the system design and implementation scale than in introducing fundamentally new RL ideas or algorithms.
>
> We agree with your high-level point: our paper focuses on identifying the right RL techniques and adapting them to the GUI-agent training, rather than proposing entirely new RL algorithms in the abstract. In the process, we introduced several non-trivial, domain-driven innovations that are essential for making RLVR feasible and effective in mobile GUI setting.
>
> 1. SPA reward
>
> Prior work on length penalties for LLMs—such as LASER, LACONIC, and adaptive length shaping methods—operates on token sequences, where length is directly tied to text generation cost or conciseness. In contrast, our SPA reward operate on multi-step GUI action trajectories.
>
> Instead of token penalizing raw length, SPA only shapes successful trajectories, modulated by the shortest known successful path within each task group. This design prevents both “looping explorations” and “early-stop failures”. Thus, while inspired by length-aware shaping, SPA solves a qualitatively different problem in a way that existing formulations cannot be directly applied to.
>
> 2. FCF
>
> Recent LLM curriculum learning  methods such as Difficulty-targeted Data Selection focus on difficulty-based reprioritization. These strategies assume that most tasks are solvable given enough samples and that exploration cost is relatively cheap. In GUI-agent RL, the situation is inverted: Many tasks exhibit heavy-tailed difficulty and interaction steps are extremely expensive.
>
> Our Failure Curriculum Filtering (FCF) is a simple but effective resource-aware filtering mechanism. It tracks tasks that remain unsolved, temporarily cools them down to avoid wasted sampling, and shifts updates toward tasks where the current policy can make progress. This produces an online, failure-driven curriculum aligned with the cost structure of GUI environments.
>
> 3. AdaPR
>
> Prioritized replay methods such as PER and distributed variants like DPER and TBA rely on dense or multi-step reward signals that make TD-error or recency informative. The task distribution exhibits heavy-tailed difficulty, which means that successful trajectories—especially on less frequently solved tasks—are rare and thus should be reused efficiently.
>
>
> With these constraints in mind, AdaPR is designed not as a replacement for existing replay methods, but as a domain-adapted instantiation of prioritized replay that maximizes the utility of scarce, informative trajectories. Concretely:
>
> - it maintains a replay buffer of successful, high-advantage trajectories, which are particularly meaningful because a high advantage in our setting indicates that successful rollouts for the same task are relatively rare under the current policy.
>
> - it up-weights rare successes on difficult commands to ensure that their learning value is not diluted by the dominance of easy or failed rollouts;
>
> - it continuously mixes replayed successes with fresh on-policy data, providing stability during GRPO optimization.
>
> Rather than aiming to outperform or contradict existing replay formulations, AdaPR addresses the cost–difficulty profile of GUI agents, ensuring that each expensive interaction step is used to its effect.
>
> Together, they create a training regime that optimize GRPO under sparse binary rewards. This integration is what makes large-scale GUI-agent RLVR tractable.
>
> We provide a more detailed comparison with prior related algorithms in the revised version, specifically in Appendix F: Extended Background on Prior Approaches.

---

> > ### Author Response · Authors · 2025-11-21
> > **Response to Reviewer TUvD (2/2)**
> >
> > > Q2: I have some reservations about the mixture sampling strategy. The historical “successful” trajectories stored in the buffer are highly off-policy, while the underlying training objective (a clipped PPO-style GRPO variant) is fundamentally on-policy. As ADAGRPO employs a clipping mechanism, these off-policy samples may often fall outside the acceptable distribution range and thus get heavily clipped, thereby limiting their effective gradient contribution. This raises a question about sample efficiency and the real utility of replayed data. It would be helpful if the authors could provide some experiments to show the clip ratio of the off-policy samples or effective sample contribution.
> >
> >
> >
> > Thank you for raising this important question regarding whether replayed trajectories—being off-policy—might fall outside the PPO-style clipping range and thus contribute little to the gradient.
> >
> > We provide additional analyses of the AdaPR buffer in Appendix B.3.3 Effectiveness of the AdaPR Buffer, including its entry and exit criteria, clipping behavior, and its contribution to actual training dynamics. Below is a summary of the added content; the corresponding figures can be found in the revised appendix.
> >
> > 1. Strict buffer design limits off-policy drift.
> > Only the top-κ trajectories (κ = 25%) enter the replay buffer, and eviction follows two rules:
> > (1) trajectories older than 4 rounds are removed, and
> > (2) if full, the buffer drops the lowest-advantage trajectories.
> > This ensures replayed trajectories come from *recent* policies and remain close to the current distribution.
> >
> > 2. Clip ratio analysis shows replay samples are mostly unclipped.
> > Figure 11 reports the token-level clip ratio.
> > Before enabling AdaPR, the clip ratio is 0.5–1%. After enabling AdaPR, it briefly peaks at ~6% and stabilizes at 3–4%.
> > Given that batches contain roughly 50% replay samples, this implies >90% of replayed tokens lie within the unclipped region and thus contribute fully to the gradient.
> >
> > 3. Replay maintains broad task coverage.
> > Figure 12 shows that after warm-up, the number of unique tasks from on-policy and replay samples remains ~1:1, indicating that the buffer retains diverse, up-to-date successful behaviors.
> >
> > 4. Replay improves trajectory-level accuracy.
> > As shown in Figure 13, adding replay improves trajectory-level success by over 5% absolute, demonstrating that replayed samples are not only valid but beneficial for learning.
> >
> > 5. Replay significantly improves model capability.
> > In the ablation study shown in Figure 3(b), we compare the trajectory-level training rewards of MobileRL and MobileRL w/o AdaPR. Note that this curve includes only on-policy samples; we intentionally exclude off-policy samples from the buffer, as most of those trajectories are already successful and would obscure the comparison. Even under this stricter evaluation, AdaPR leads to a substantial reward improvement starting from training step ~10 and also prevents the mid-training collapse observed in the no-replay variant.
> >
> >
> > These analyses confirm that replayed trajectories are only mildly off-policy, remain mostly inside the clipping range, and provide significant effective gradient signal. AdaGRPO thus maintains strong sample efficiency while leveraging successful past rollouts.
> >
> > ---
> >
> > > Q3: There is no training recipe in the provided codebase, which is a core component of the proposed framework. Did I miss it?
> >
> > We plan to open-source this part of the code after the paper is accepted. Nevertheless, we have provided a copy of the training codebase in the training folder at https://anonymous.4open.science/r/MobileRL-iclr-4513
> > so that you can inspect the detailed implementation of the functionalities described in the paper.

---

> ### Author Response · Authors · 2025-11-27
> **Thanks for your great review!**
>
> Dear Reviewer,
>
> Thank you very much again for your kind and valuable reviews! We hope our previous response addresses your questions, from which this work has benefited a lot. If possible, we would be deeply grateful if you could help share suggestions to further make MobileRL a stronger work!
>
> Thank you again for your encouraging comments!

---

### Official Review · Reviewer_Q2uz · 2025-10-30

**Soundness:** 4
**Presentation:** 4
**Contribution:** 3
**Rating:** 8
**Confidence:** 5

**Summary:**

The authors introduce MobileRL, an online agentic reinforcement learning framework for mobile GUI control. They highlight large-scale environmental interaction and heavy-tailed task difficulty as current challenges in scaling up RL for GUI control. To address this, they build on top of GRPO with three main components: (1) off-policy sampling from a buffer of previously successful trajectories, (2) a curriculum that downweighs and filters out low-success rate tasks, and (3) reward shaping to encourage a shortest-path solution. After performing SFT using reasoning and non-reasoning trajectories, they apply MobileRL to achieve, at the time of writing, the state of the art in native model (non-agent-system-based) GUI control.

**Strengths:**

Originality: To the best of my knowledge, prior work has not applied oversampling of successful trajectories, reward shaping to encourage shortest paths, and curriculum learning to the GUI and multi-turn agent-LLM setting. The reward shaping is particularly interesting, since encouraging shortest paths has been underexplored in this setting, and the specific formulation is a good contribution.

Quality: The work is of high quality. The core contributions are sufficiently explained, and the experimental analysis is comprehensive and supportive.

Clarity: The work is generally very clear.

Significance: Given the rapid extension of LLMs to agentic and multi-turn settings, and the growing interest in the GUI setting, this work is very significant. This is especially evident in Android World, which, at the time of writing, is SOTA for a native model (non-agentic system). Open-sourcing the RL codebase is also a significant contribution to open research practices, although the current link is dead; this should be rectified in the rebuttal.

**Weaknesses:**

Originality: It could be made more evident how the specific components have been explored in other domains using LLMs, RL, and multi-turn interaction. For example, it is now commonplace to avoid vanishing learning signal through sampling strategies (e.g. DAPO). Similarly, curriculum learning and reward shaping exist in other non-LLM RL settings. It would be helpful to clarify whether the specific contributions, such as the SPA reward, are present in different settings.

Quality: Some experiments and conclusions may be slightly misleading, possibly due to a lack of clarity at points, as I explain below.

Clarity: Some aspects seek clarification, which I expand on in the questions section below.

Significance: While the Android World results are significant, I encourage the authors to clarify that it is SOTA for native agents, but not all agentic systems (see the current leaderboard.

**Questions:**

1.	The problem formulation (line 150) implies that the agent is provided with no form of memory, only the most recent screenshot. The results seem surprising for this setting. Can the authors please clarify that this is correct?
2.	In line 225, could you please clarify whether the SPA reward is still broadcast to every timestep?
3.	In Figure 3b, are the trajectory-level rewards before or after the SPA penalty has been applied?
4.	In Figure 3b, could the authors provide some intuition on why the MobileRL w/o SPA & AdaPR collapses? Specifically, why does only the combination of these two components prevent collapse? Intuitively, introducing oversampling of positive trajectories would avert collapse. But it is less intuitive that the SPA reward (in isolation from oversampling) would prevent collapse, since this reward changes the learning problem but not necessarily the dynamics.
5.	Can the authors please clarify whether difficult tasks are only removed for training, but still included for testing? Naturally, there would be instant gains in reward from removing tasks.
6.	Could the authors please clarify whether MobileRL w/o FCF includes AdaPR or SPA?
7.	It is unsurprising that the reward ceiling for MobileRL w/o FCF is lower, since low-reward tasks have been removed. However, why does the curve in Figure 3b start so much lower? It is my understanding that tasks are removed throughout training after at least two epochs (line 260), which suggests that all lines in the figure should start with the same reward.

---

> ### Author Response · Authors · 2025-11-21
> **Response to Reviewer Q2uz (1/2)**
>
> Thank you very much for your constructive suggestions and for your appreciation of our work. Below, we provide point-by-point responses to further clarify our contributions.
>
> > Q1: The problem formulation (line 150) implies that the agent is provided with no form of memory, only the most recent screenshot. The results seem surprising for this setting. Can the authors please clarify that this is correct?
>
> We do not incorporate any explicit memory module. However, this does not mean the agent has no access to prior context. Our definition of the current state includes the history of interactions—meaning the agent can infer what has happened based on previous `think` and `action`. What the agent does not receive is the full history of screenshots or XML states. Including these would make the context prohibitively long (a single screenshot + XML already occupies ~6k tokens).
>
> ---
>
> > Q2: In line 225, could you please clarify whether the SPA reward is still broadcast to every timestep?
>
> Yes. In our setting, the SPA reward continues to be broadcast to every timestep.
>
> ---
>
> > Q3: In Figure 3b, are the trajectory-level rewards before or after the SPA penalty has been applied?
>
> They are measured before the SPA penalty is applied. You can interpret the results in the figure as indicating only whether the sampled trajectory from the training set was executed correctly in that round (a binary 0/1 reward). Trajectories from the AdaPR buffer are excluded because they are almost always successful; including them would skew the curve and offer less meaningful insight than focusing solely on trajectories sampled from the training set.
>
> ---
>
> > Q4: In Figure 3b, could the authors provide some intuition on why the MobileRL w/o SPA & AdaPR variant collapses? Specifically, why does only the combination of these two components prevent collapse? Intuitively, oversampling positive trajectories should avert collapse. But it is less intuitive that the SPA reward—without oversampling—would prevent collapse, since this reward changes the optimization problem but not necessarily the data dynamics.
>
> Your interpretation of AdaPR is correct: oversampling successful trajectories can indeed prevent collapse, and we observe this effect empirically.
>
> Our intuition for SPA is as follows. Without SPA, the agent can initially learn reasonable behaviors, but the length of successful action sequences increases. Because rewards without SPA are assigned to individual steps rather than whole trajectories, longer correct sequences end up accumulating more positive reward.
>
> This increase in sequence length creates two problems:
>
> - The SFT stage did not include sufficiently long operation sequences (e.g., those exceeding 50 steps), so the model’s ability to handle very long action histories is limited.
>
> - The benchmark itself imposes strict limits on rollout length—for example, AndroidWorld restricts rollouts to at most twice the length of the human demonstration and forcibly terminates any episode that exceeds this limit.
>
> As a result, the w/o SPA variant fails to complete these longer (and typically more difficult) tasks. Under the FCF mechanism, such long and difficult tasks are sampled less frequently and eventually filtered out.
>
> ---
>
> > Q5: Can the authors please clarify whether difficult tasks are only removed for training, but still included for testing? Removing tasks would give an immediate gain in reward.
>
> Yes. Difficult tasks are filtered only in training set. Every evaluation uses the full test set without removing any tasks.
>
> ---
>
> > Q6: Could the authors please clarify whether MobileRL w/o FCF includes AdaPR or SPA?
>
> MobileRL w/o FCF includes both AdaPR and SPA.
>
> ---
>
> > Q7: It is unsurprising that the reward ceiling for MobileRL w/o FCF is lower, since low-reward tasks have been removed. However, why does the curve in Figure 3b start so much lower? Line 260 suggests that tasks are removed only after at least two epochs, so all curves should start at similar reward levels.
>
> Please refer to Section 2.2.3 Failure Curriculum Filtering (FCF) (lines 264–266): to accelerate convergence to stable training, our FCF variant retains historical failure information. As a result, even in the earliest training rounds, some tasks already have reduced sampling probability. This practical design choice leads to the MobileRL w/o FCF curve beginning at a different reward level in Figure 3b.

---

> > ### Author Response · Authors · 2025-11-21
> > **Response to Reviewer Q2uz (2/2)**
> >
> > > Weakness: Originality: It could be made more evident how the specific components have been explored in other domains using LLMs, RL, and multi-turn interaction. For example, it is now commonplace to avoid vanishing learning signal through sampling strategies (e.g. DAPO). Similarly, curriculum learning and reward shaping exist in other non-LLM RL settings. It would be helpful to clarify whether the specific contributions, such as the SPA reward, are present in different settings.
> >
> > We provide a more detailed comparison with prior related algorithms in the revised version, specifically in Appendix F: Extended Background on Prior Approaches. Here is a summary of our specific contributions.
> >
> > 1. SPA reward
> >
> > Prior work on length penalties for LLMs—such as LASER, LACONIC, and adaptive length shaping methods—operates on token sequences, where length is directly tied to text generation cost or conciseness. In contrast, our SPA reward operate on multi-step GUI action trajectories.
> >
> > Instead of token penalizing raw length, SPA only shapes successful trajectories, modulated by the shortest known successful path within each task group. This design prevents both “looping explorations” and “early-stop failures”. Thus, while inspired by length-aware shaping, SPA solves a qualitatively different problem in a way that existing formulations cannot be directly applied to.
> >
> > 2. FCF
> >
> > Recent LLM curriculum learning  methods such as Difficulty-targeted Data Selection focus on difficulty-based reprioritization. These strategies assume that most tasks are solvable given enough samples and that exploration cost is relatively cheap. In GUI-agent RL, the situation is inverted: Many tasks exhibit heavy-tailed difficulty and interaction steps are extremely expensive.
> >
> > Our Failure Curriculum Filtering (FCF) is a simple but effective resource-aware filtering mechanism. It tracks tasks that remain unsolved, temporarily cools them down to avoid wasted sampling, and shifts updates toward tasks where the current policy can make progress. This produces an online, failure-driven curriculum aligned with the cost structure of GUI environments.
> >
> > 3. AdaPR
> >
> > Prioritized replay methods such as PER and distributed variants like DPER and TBA rely on dense or multi-step reward signals that make TD-error or recency informative. The task distribution exhibits heavy-tailed difficulty, which means that successful trajectories—especially on less frequently solved tasks—are rare and thus should be reused efficiently.
> >
> >
> > With these constraints in mind, AdaPR is designed not as a replacement for existing replay methods, but as a domain-adapted instantiation of prioritized replay that maximizes the utility of scarce, informative trajectories. Concretely:
> >
> > - it maintains a replay buffer of successful, high-advantage trajectories, which are particularly meaningful because a high advantage in our setting indicates that successful rollouts for the same task are relatively rare under the current policy.
> >
> > - it up-weights rare successes on difficult commands to ensure that their learning value is not diluted by the dominance of easy or failed rollouts;
> >
> > - it continuously mixes replayed successes with fresh on-policy data, providing stability during GRPO optimization.
> >
> > Rather than aiming to outperform or contradict existing replay formulations, AdaPR addresses the cost–difficulty profile of GUI agents, ensuring that each expensive interaction step is used to its effect.
> >
> > Together, they create a training regime that optimize GRPO under sparse binary rewards. This integration is what makes large-scale GUI-agent RLVR tractable.
> >
> >
> >
> > ---
> >
> > > Weakness: Significance: While the Android World results are significant, I encourage the authors to clarify that it is SOTA for native agents, but not all agentic systems see the current leaderboard.
> >
> > Due to the requirements of anonymous review, I’m unable to respond to this request. We will update the correct benchmark description once the paper is made public.

---

> ### Author Response · Authors · 2025-11-27
> **Thanks for your great review!**
>
> Dear Reviewer,
>
> Thank you very much again for your kind and valuable reviews! We hope our previous response addresses your questions, from which this work has benefited a lot. If possible, we would be deeply grateful if you could help share suggestions to further make MobileRL a stronger work!
>
> Thank you again for your encouraging comments!

---

### Official Review · Reviewer_ybyQ · 2025-10-30

**Soundness:** 3
**Presentation:** 3
**Contribution:** 3
**Rating:** 4
**Confidence:** 4

**Summary:**

This paper proposes an online agentic reinforcement learning framework named MOBILERL, designed to enhance mobile GUI agents. The framework aims to solve challenges in developing GUI agents with RL, such as the heavy-tailed distribution of task difficulty and inefficient large-scale environment sampling. The core of MOBILERL is its Difficulty-ADAptive GRPO (ADAGRPO) algorithm. This algorithm stabilizes training and improves sample efficiency through three strategies including Difficulty-Adaptive Positive Replay (AdaPR), Failure Curriculum Filtering (FCF), and Shortest-Path Reward Adjustment (SPA).

**Strengths:**

ADAGRPO improves training stability and efficiency through three key strategies : 1) Difficulty-Adaptive Positive Replay (AdaPR) to amplify the learning signal from rare, informative successful trajectories; 2) Failure Curriculum Filtering (FCF) to optimize computational budget by dynamically down-weighting or removing persistently unsolvable tasks ; and 3) Shortest-Path Reward Adjustment (SPA), which uses reward shaping to encourage more efficient (shorter) solutions. The framework's effectiveness is thoroughly validated on two major benchmarks, AndroidWorld and AndroidLab, where the MOBILERL-9B model achieves state-of-the-art success rates of 80.2% and 53.6%, respectively, significantly outperforming previous results.

**Weaknesses:**

1. I question the generality of the MobileRL-9B model's SOTA results. Appendix F and D.1 state the best 9B model on AndroidWorld was trained "purely on the AndroidWorld environment". This raises doubts about whether the 53.6% score on AndroidLab in Table 1  came from the same model. Attributing SOTA scores from different training strategies to a single model name (MobileRL-9B) may be inconsistent.

2. there is no testing on "out-of-domain" public-use apps.

**Questions:**

1. Given that MobileRL uses both images and XML as input, did the baseline models also utilize XML?

2. Appendices F and D.1 seem to indicate that the 80.2% score on AndroidWorld was achieved by a model trained only on that dataset. Does this mean the 53.6% score on AndroidLab came from a separate model using a different training strategy?

3. The FCF strategy permanently removes tasks with consecutive failures from the training pool. Could this distort the training objective, leading the model to achieve a high success rate simply by "giving up" on difficult tasks and optimizing for the remaining, simpler task distribution?

---

> ### Author Response · Authors · 2025-11-21
> **Response to Reviewer ybyQ (1/2)**
>
> Thank you very much for your constructive suggestions and for your appreciation of our work. Below, we provide point-by-point responses to further clarify our contributions.
>
>
> > Q1:I question the generality of the MobileRL-9B model's SOTA results. Appendix F and D.1 state the best 9B model on AndroidWorld was trained "purely on the AndroidWorld environment". This raises doubts about whether the 53.6% score on AndroidLab in Table 1 came from the same model. Attributing SOTA scores from different training strategies to a single model name (MobileRL-9B) may be inconsistent.
>
> > Appendices F and D.1 seem to indicate that the 80.2% score on AndroidWorld was achieved by a model trained only on that dataset. Does this mean the 53.6% score on AndroidLab came from a separate model using a different training strategy?
>
> We guarantee that both reported results (80.2% on AndroidWorld and 53.6% on AndroidLab) come from the same fully trained agent checkpoint (MobileRL-9B). To verify this, we have uploaded the final checkpoint in the https://huggingface.co/anonymous-c58sihud455eafaewdf/MobileRL-review, and you can reproduce both results by following the docker-based deployment instructions.
>
> The misunderstanding arises from interpreting Appendices F and D.1 as implying that the model was trained only on the AndroidWorld dataset. While the RL stage indeed did not use the AndroidLab training set, the two SFT stages incorporated extensive human-annotated data drawn from a mixture of open-source and self-constructed task sets—including the official AndroidLab training set. Section 3.3 (Ablation Study) further shows that MobileRL-9B’s improvements on AndroidLab primarily originate from these SFT stages. From another perspective, because we unify the state representation and action space across different mobile benchmarks, the operational logic learned during RL training on AndroidWorld can transfer seamlessly to AndroidLab. Although the two benchmarks differ in apps, evaluation metrics, and task compositions, both define natural-language mobile tasks that reflect common real-world phone interactions, resulting in a meaningful degree of similarity. Although we did not explicitly claim cross-benchmark generalization, the consistency across benchmarks suggests that the model indeed benefits from such generalization.
>
> ---
>
> > Q2: Given that MobileRL uses both images and XML as input, did the baseline models also utilize XML?
>
>
> The baseline models vary in whether they use XML. Our general principle is:
>
> 1. Prefer the highest officially reported score.
> 2. If not available, replicate testing using the model’s recommended phone-use format.
> 3. Otherwise, evaluate using our pipeline.
>
> Below is the detailed input configuration for each baseline:
>
> - GPT-4o-2024-11-20: screenshot-only input.
> - Claude-Sonnet-4-20250514-thinking, GLM-4.1V-9B-Thinking: no official phone-operation template available; we evaluated them using our input format (screenshot + XML).
> - UI-Tars-1.5, UI-Tars-7B, Qwen2.5-VL-7B-Instruct: evaluated using their official operation templates, adapted to our environment (action space, input format, system prompt). State input was screenshot-based.
> - AutoGLM-2024-10, UI-Genie-Agent: used results reported in their papers (screenshot + XML).
> - V-Droid: used official results; text-only model relying on XML-based representations.
>
> It should be explicitly noted that for models extensively trained for GUI interaction (e.g., UI-Tars), deviating from their prescribed input format would artificially lower their performance. Therefore, for such models, omitting XML is a fairer comparison.

---

> > ### Author Response · Authors · 2025-11-21
> > **Response to Reviewer ybyQ (2/2)**
> >
> > > Q3: The FCF strategy permanently removes tasks with consecutive failures from the training pool. Could this distort the training objective, leading the model to achieve a high success rate simply by "giving up" on difficult tasks and optimizing for the remaining, simpler task distribution?
> >
> >
> > While removing tasks with consecutive failures does shift sampling toward more solvable tasks during training, our ablation studies show that this does not inflate performance through “giving up” on difficult tasks.
> >
> > - Without FCF, the score drops to 64.8, which is 6.3 points lower (Fig. 3c).
> > - On difficulty levels 3 & 4, FCF yields over +10% pass@1 improvement (Fig. 4).
> > - In Appendix E.2 Difficulty-Aware Evaluation on AndroidWorld, we conduct additional experiments on difficulty grading, all of which show that our method is equally effective at increasing performance on challenging problems. In particular, for high-difficulty tasks containing the multi-goal tag, we observe a pass@1 improvement from 0.042 to 0.317 before and after RL, further reinforcing this point.
> >
> > FCF is designed for the 100-step training-budget constraint. Given our hardware limitations—each experiment exceeds 40 hours—we cannot fully explore significantly longer training horizons. Under constrained resources, FCF helps focus sampling and training on tasks where improvement is most feasible, thereby increasing overall training efficiency without reducing difficulty coverage.
> >
> > ---
> >
> > > Weakness: there is no testing on "out-of-domain" public-use apps
> >
> >
> > To evaluate generalization to real-world public-use apps outside AndroidWorld and AndroidLab, we constructed a new out-of-domain evaluation suite.
> >
> > We use 21 widely used public Android apps, including:
> > Settings, Booking, Chrome, Contacts, Expedia, Gmail, Google Drive, GoogleCalendar, GoogleDocs, GoogleKeep, GoogleMaps, GooglePlayBooks, GooglePlayStore, Joplin, OsmAnd, Quora, Reddit, TikTok, Twitter (X), Temu, Broccoli.
> >
> > We created 106 natural-language tasks covering realistic user goals, such as:
> > - Web navigation and incognito mode in Chrome
> > - Email composition in Gmail
> > - Navigation queries in Google Maps
> > - Posting / liking / bookmarking in TikTok or Twitter (X)
> > - Search / sort / cart manipulation in Temu
> >
> > Each trajectory is labeled with {0, 0.5, 1} by human annotation:
> > - 0: incorrect / goal not achieved
> > - 0.5: partially correct with undesirable redundancy
> > - 1: correct with no unacceptable redundancy
> >
> > The final score is the average over 106 tasks.
> >
> > We evaluated two models: MobileRL-9B w/o ADAGRPO and MobileRL-9B (full method). As shown in table below, ADAGRPO improves overall score by 2.7 points. These results demonstrate that the RL training scheme (including ADAGRPO) improves not only in-domain benchmarks, but also generalization to entirely unseen, real-world public apps. Full details are in Appendix E.1 Out-of-domain Evaluation on Public-use Android Applications.
> >
> >
> > | Model | 0 | 0.5 | 1 | Overall |
> > |-------|------|--------|------|---------|
> > | MobileRL-9B w/o ADAGRPO | 0.4476 | 0.0392 | 0.5132 | 0.5328 |
> > | MobileRL-9B | 0.4225 | 0.0367 | 0.5409 | 0.5592 |
> >
> >
> > The complete set of task prompts, as well as the detailed behaviors of the two agents on each app, are provided in the human-evaluation folder at https://anonymous.4open.science/r/MobileRL-iclr-4513.

---

> > > ### Comment · Reviewer_ybyQ · 2025-11-23
> > >
> > > Thank the authors for their response and detailed clarifications.
> > >
> > > The method strictly relies on XML for grounding (limiting use on apps like TikTok). How does the model behave when XML is partially missing, misaligned, or unavailable? or Did you conduct experiments using "Vision-only" or "XML-only" inputs?

---

> > > > ### Author Response · Authors · 2025-11-24
> > > > **Response to Reviewer ybyQ regarding XML and Vision-only question**
> > > >
> > > > This is an excellent point: XML obviously cannot cover all possible clickable areas. Therefore, our team has been actively exploring a full transition toward Vision-only, i.e., removing XML from the state representation. We answer your question in two parts:
> > > >
> > > > 1. How the current model behaves when XML is missing, and
> > > > 2. The results of re-running the entire pipeline in Vision-only mode.
> > > >
> > > > ---
> > > >
> > > > 1. Performance of the current model when XML is entirely missing
> > > >
> > > > First, it is important to clarify that the current model’s full training pipeline uses both screenshot + XML as the state input. Removing XML entirely will certainly cause significant performance degradation. However, our test set does include cases similar to the TikTok scenario you mentioned, where the target clickable area is not present in the XML. Below are two such cases:
> > > >
> > > >
> > > >
> > > > Case 1
> > > >
> > > > Task: Create a note in Markor named kind_mouse_backup.md, paste some content into the note, and save it.
> > > >
> > > > During execution, the paste operation corresponds to an element that does not appear in XML. Nevertheless, the model successfully predicted a corresponding bounding box directly from the image. In `contacts_3-491638-1757064623218-step7.jpg`, all XML-derived bounding boxes are shown in green, and the predicted box is shown in red with a red cross indicating the actual ADB click location.
> > > >
> > > > It shows that our agent predicted a correct location for the paste operation, even though it was absent from XML.
> > > >
> > > > We provide the full execution log and visualization in the supplementary material under `MarkorCreateNoteFromClipboard_1758557688435`.
> > > >
> > > >
> > > >
> > > > Case 2
> > > >
> > > > Task: Add a contact named Xu, set the work phone number to 12345678 and the mobile number to 87654321.
> > > >
> > > > Although our agent did not ultimately execute this task completely correctly, one step requires selecting the “work” label from a dropdown menu. However, the dropdown contents do not appear in the XML. In `contacts_3-491638-1757064623218-step7.jpg`, we again plot all XML boxes and the model prediction. The agent predicted the correct location for the “work” label even though it is not present in the XML （both label and bbox are not in XML).
> > > >
> > > > We provide the full log and visualization in the supplementary material under `contacts_3_1757064623219`.
> > > >
> > > >
> > > >
> > > > Although RL training enables the agent to correctly ground actions directly from images in these cases, we acknowledge that our current method does not fully address scenarios where XML is severely missing. This is because most academic benchmarks evaluated in the paper contain complete and clean XML.
> > > >
> > > > To further investigate this, we simulated the XML-unavailable setting by replacing the model input with screenshot-only, and evaluated MobileRL-9B on the AndroidWorld test set.
> > > > Under this extreme condition, MobileRL-9B achieves a score of 24.6%.
> > > >
> > > > Despite the mismatch with the training format, the agent demonstrates non-trivial grounding capability from screenshots alone.
> > > >
> > > > ---
> > > >
> > > > 2. Re-running the entire pipeline in Vision-only mode
> > > >
> > > > We further re-trained the entire pipeline (Reasoning-Free SFT, Reasoning SFT, and RL) in a pure Vision-only setting. We found that—with current model capacity—agents trained in this mode perform only slightly worse than those trained with screenshot+XML. The major differences in the training setup are:
> > > >
> > > > - All state representations and history messages exclude XML; system prompts are updated accordingly.
> > > > - All element bounding boxes (from XML) are replaced with points predicted directly from vision, covering Tap, Swipe, etc.
> > > > - Coordinate formats are converted to absolute (for Qwen2.5-VL) or relative (for GLM4V), based on the model’s pretrained representation.
> > > >
> > > > Other parts of the training pipeline remain unchanged.
> > > >
> > > > Using GLM-4.1V-9B-Base as the base model, we obtain the following results:
> > > >
> > > > | Method               | Screenshot-only | Screenshot+XML |
> > > > | -------------------- | --------------- | -------------- |
> > > > | + Reasoning-Free SFT | 47.5            | 48.1           |
> > > > | + Reasoning SFT      | 62.6            | 66.2           |
> > > > | + AdaGRPO            | 79.3            | 80.2           |
> > > >
> > > > ---
> > > >
> > > > We did not run an XML-only baseline on the same dataset because XML-only clearly has a lower performance ceiling and is typically used only when the model’s visual grounding ability is weak.
> > > >
> > > > If you have any further questions, we would be happy to discuss!

---

> ### Author Response · Authors · 2025-11-27
> **Thanks for your great review!**
>
> Dear Reviewer,
>
> Thank you very much again for your kind and valuable reviews! We hope our previous response addresses your questions, from which this work has benefited a lot. If possible, we would be deeply grateful if you could help share suggestions to further make MobileRL a stronger work!
>
> Thank you again for your encouraging comments!

---

### Author Response · Authors · 2025-11-25
**General Response**

Dear Area Chair,

We thank you and all reviewers for your time and constructive feedback, which has greatly helped us improve our work. In our revision and rebuttal, we addressed all major concerns and added new experiments, analyses, and open-source materials.

Below, we summarize common concerns shared by multiple reviewers:

## Common Concerns Raised by Multiple Reviewers

- **Generalization beyond in-domain benchmarks**
  Reviewers asked whether gains transfer to truly unseen, real-world apps (beyond AndroidWorld/AndroidLab-like settings).
  - *Response:* We added an out-of-domain human-evaluated suite on **21 public Android apps** with **106 tasks**, where ADAGRPO improves the overall score from **0.5328 → 0.5592**.

- **Novelty vs. “re-purposing” known RL techniques**
  Reviewers asked us to better position SPA/AdaPR/FCF relative to length penalties, prioritized replay, and curriculum learning.
  - *Response:* We expanded the related-work comparison to clarify these are **domain-driven adaptations** for GUI-agent RL under **sparse binary rewards** and **expensive interaction**, rather than direct reuse.

- **Replay (AdaPR) under PPO-style clipping: effectiveness & on-policy compatibility**
  Reviewers worried replayed trajectories are too off-policy and get clipped, limiting gradient signal.
  - *Response:* We added buffer design details (recency-based eviction, advantage-based retention) and a **token-level clip-ratio analysis** showing replay is mostly **unclipped** (stabilizing around **3–4%** after enabling AdaPR).

## Reviewers’ Concerns and How We Addressed Them
We provided point-by-point clarifications and additional analyses/experiments to address all major concerns:

### Reviewer ybyQ
- **Concern: Whether AndroidWorld (80.2%) and AndroidLab (53.6%) results come from the same “MobileRL-9B” checkpoint.**
  **Response:** We confirmed both results are from the same checkpoint and uploaded it with docker-based reproducibility instructions.
- **Concern: Baseline input modality fairness (screenshot vs screenshot+XML).**
  **Response:** We specified each baseline’s input configuration, using official scores/templates when available; otherwise we evaluated in our pipeline and stated the exact format.
- **Concern: FCF might inflate performance by “giving up” on hard tasks.**
  **Response:** We clarified evaluation always uses the full test set; ablations show removing FCF reduces performance (e.g., -6.3 on AndroidWorld), and difficulty-aware results show larger gains on higher difficulty levels.
- **Weakness: Lack of out-of-domain evaluation on public-use apps.**
  **Response:** We added the above out-of-domain suite (21 apps / 106 tasks), where ADAGRPO improves **0.5328 → 0.5592** (+2.7 points).

### Reviewer Q2uz
- **Clarification: State/memory formulation.**
  **Response:** No explicit memory module; the agent receives interaction history (think/action), but not full screenshot/XML history due to context length.
- **Concern: Why “w/o SPA & AdaPR” collapses; why SPA helps.**
  **Response:** Without SPA, success trajectories lengthen, leading to rollout-limit failures and mismatch with SFT exposure to long histories; SPA encourages shorter successful paths and stabilizes training.
- **Concern: Positioning of novelty and “SOTA” statement.**
  **Response:** We added extended comparisons (Appendix F) to better contextualize SPA/FCF/AdaPR and clarify scope.

### Reviewer TUvD
- **Concern: Off-policy replay may be heavily clipped under PPO-style GRPO, limiting usefulness.**
  **Response:** We added AdaPR buffer analyses (Appendix B.3.3), including entry/eviction design and token-level clip ratios showing replay is largely unclipped (stabilizing at **3–4%**).
- **Concern: Missing training recipe in the codebase.**
  **Response:** We provided the training implementation in the “training” folder of the anonymous repository for inspection, and will fully open-source after acceptance.

### Reviewer Dgh2
- **Concern: Circularity in “complexity” definition and conflation between difficulty and solution length.**
  **Response:** We clarified AndroidWorld’s step budget is derived from human demonstrations (≈2×) and is independent of model behavior; we added multi-dimensional difficulty analysis using benchmark metadata, showing strongest RL gains on higher semantic/composite difficulty (e.g., S3 pass@1 **0.23 → 0.35**, D3 pass@1 **0.18 → 0.30**).
- **Concern: Reward model bias on AndroidLab (86% accuracy).**
  **Response:** We quantified errors on a manually labeled set (accuracy ~0.85; mainly false positives), capped AndroidLab sampling to <20% during RL, and ran a controlled experiment showing a modest ~4-point gap vs verified-reward training (with verified evaluation).
- **Concern: Out-of-domain generalization.**
  **Response:** Same as above, we added a public-app evaluation suite demonstrating improved generalization.

---

### Meta-Review · Area_Chair_VQT5 · 2026-01-07

**Summary:**

The paper introduces MobileRL, an online reinforcement learning framework for mobile GUI agents. The framework utilizes a Difficulty-Adaptive GRPO (ADAGRPO) algorithm which includes three key components: Difficulty-Adaptive Positive Replay (AdaPR), Failure Curriculum Filtering (FCF), and Shortest-Path Reward Adjustment (SPA). These mechanisms aim to address challenges such as sparse rewards, heavy-tailed task difficulty, and sampling inefficiency. The authors validate their method on AndroidWorld and AndroidLab benchmarks, achieving state-of-the-art results with their MobileRL-9B model.

The paper presents a solid system paper that pushes the state-of-the-art in mobile GUI agents. While the algorithmic components are adaptations of known techniques, their integration and tuning for this specific, high-value domain constitute a significant contribution. The empirical results are strong and well-supported by ablations. The authors effectively addressed the major concerns regarding generalization and technical correctness. The concerns of the negative reviewer (Dgh2) regarding circularity appear to be based on a misunderstanding of the benchmark's definitions, which the authors clarified. The skepticism of reviewer ybyQ regarding the consistency of results was directly refuted by the authors' confirmation and reproducibility materials.

**Reviewer Concerns:**

Addressed Concerns:

Methodological Consistency (Reviewer ybyQ): The concern that results on different benchmarks might come from different models was definitively refuted. The authors confirmed a single checkpoint was used and provided reproducibility instructions.

Evaluation Rigor (Reviewer ybyQ): The worry that the Failure Curriculum Filtering (FCF) might "game" the metrics by removing hard tasks was resolved by clarifying that filtering only occurs during training; the full test set is used for evaluation.

Generalization (Reviewers ybyQ, Dgh2): The request for out-of-domain evaluation was met with a new experiment on 21 public apps, demonstrating positive transfer.

Technical Validity of Replay (Reviewer TUvD): The concern that PPO clipping would render off-policy replay data ineffective was addressed with a specific analysis showing clip ratios stabilize at low levels (3-4%), confirming the data's utility.

Benchmark Definitions (Reviewer Dgh2): The reviewer's claim that task complexity was defined circularly was shown to be factually incorrect (complexity is derived from human baselines).

Input Modality (Reviewer ybyQ): Concerns about XML dependency were addressed by analyzing vision-only performance.

Outstanding Concerns:

Incremental Novelty (Reviewers TUvD, Q2uz): While the authors justified their design choices, the critique that the individual algorithmic components (prioritized replay, curriculum learning) are adaptations of existing techniques rather than fundamental inventions remains a valid perspective on the type of contribution (system/application vs. algorithmic theory), though it does not diminish the paper's effectiveness.

Reward Model Bias (Reviewer Dgh2): While the authors quantified the impact of the reward model's noise (~4% performance gap), the reliance on an imperfect proxy reward for the AndroidLab portion of training remains an inherent limitation of the setting.

**Reviewer Scores:**

Reviewer Q2uz (Score: 8; likely no change): This reviewer was already strongly supportive, rating the paper 8/10. The authors satisfactorily answered their clarifying questions regarding memory mechanisms and reward shaping. It is highly likely Q2uz's score would have remained a strong Accept.

Reviewer ybyQ (Score: 4 -> 6 likely): This reviewer's primary reservation was a suspicion that the results on different benchmarks came from different models ("inconsistent"). The authors definitively refuted this by confirming a single checkpoint was used and providing reproducibility instructions. They also addressed the concern about the FCF strategy "gaming" the metrics. With these fundamental misunderstandings resolved, the score would likely have increased to a Weak Accept or Accept.

Reviewer TUvD (Score: 4 -> 6 likely): The main technical objection was whether the off-policy replay data would be rendered useless by PPO clipping. The authors provided a specific token-level clip ratio analysis (stabilizing at 3-4%) demonstrating that the data remains effective. This direct quantitative answer to the reviewer's hypothesis typically warrants a score increase.

Reviewer Dgh2 (Score: 2; likely mo major change): This reviewer based a significant portion of their critique on the factual error that task complexity was defined circularly by the agent's own performance. The authors clarified that complexity is defined by human baselines. While the reviewer might have acknowledged the correction, their skepticism regarding reward model bias might have kept the score low, though the rebuttal arguments significantly weakened the basis for rejection.

---

### Decision · Program_Chairs · 2026-01-26

Accept (Poster)